# HYPERDQN: A RANDOMIZED EXPLORATION METHOD FOR DEEP REINFORCEMENT LEARNING

**Ziniu Li[1], Yingru Li[1†], Yushun Zhang[1], Tong Zhang[2†], and Zhi-Quan Luo[1†]**
[1]Shenzhen Research Institute of Big Data,
The Chinese University of Hong Kong, Shenzhen, China
[2]Hong Kong University of Science and Technology
`{ziniuli, yingruli, yushunzhang}@link.cuhk.edu.cn,`
`tongzhang@ust.hk, luozq@cuhk.edu.cn`

## ABSTRACT

Randomized least-square value iteration (RLSVI) is a provably efficient exploration method. However, it is limited to the case where (1) a good feature is known in advance and (2) this feature is fixed during the training. If otherwise, RLSVI suffers an unbearable computational burden to obtain the posterior samples. In this work, we present a practical algorithm named *HyperDQN* to address the above issues under deep RL. In addition to a non-linear neural network (i.e., base model) that predicts $Q$-values, our method employs a probabilistic hypermodel (i.e., meta model), which outputs the parameter of the base model. When both models are jointly optimized under a specifically designed objective, three purposes can be achieved. First, the hypermodel can generate approximate posterior samples regarding the parameter of the $Q$-value function. As a result, diverse $Q$-value functions are sampled to select exploratory action sequences. This retains the punchline of RLSVI for efficient exploration. Second, a good feature is learned to approximate $Q$-value functions. This addresses limitation (1). Third, the posterior samples of the $Q$-value function can be obtained in a more efficient way than the existing methods, and the changing feature does not affect the efficiency. This deals with limitation (2). On the Atari suite, *HyperDQN* with 20M frames outperforms DQN with 200M frames in terms of the maximum human-normalized score. For SuperMarioBros, HyperDQN outperforms several exploration bonus and randomized exploration methods on 5 out of 9 games.

## 1 INTRODUCTION

Reinforcement learning (RL) (Sutton & Barto, 2018) involves an agent that interacts with an unknown environment to maximize cumulative reward. The trade-off between exploration and exploitation is a fundamental problem in RL (Kakade, 2003). On the one hand, the agent needs to explore highly uncertain states and actions, which may sacrifice immediate reward. On the other hand, in the long term, the agent should take the best-known action; however, this action may be sub-optimal due to partial information. To this end, sample-efficient RL agents should qualify the epistemic uncertainty (i.e., subjective uncertainty due to limited samples (Osband, 2016a)) to address the trade-off.

Bayesian approaches like Thompson sampling (Russo et al., 2018) provide a nice way of encoding the epistemic uncertainty by posterior distribution. For instance, randomized least-square value iteration (RLSVI)[1] (Osband et al., 2016b; 2019) is a well-known Bayesian algorithm. Specifically, RLSVI takes three steps to address the exploration and exploitation trade-off. First, conditioned on observed data, RLSVI solves a Bayesian linear regression problem and updates the posterior distribution over (the parameter of) the optimal $Q$-value functions by the Bayes update rule. Second, RLSVI samples a specific $Q$-value function from the posterior distribution. Third, to collect more data, greedy actions are selected based on this $Q$-value function. In theory, the randomness in posterior sampling could yield positive bias, which boosts optimistic behaviors (Osband et al., 2019; Russo, 2019; Zanette et al., 2020). However, RLSVI is restricted to the following case.

---

† : Corresponding authors.

[1]For readers who are not familiar with RLSVI, please refer to Appendix A.1 for a brief introduction.

- A good feature is known in advance. Over this feature, the $Q$-values can be approximated as a linear function.

- This feature is fixed so that the posterior distribution is easy to compute in an incremental update way (see Remark 2 in Appendix A.1 or (1.1) for an illustration).

Unfortunately, these two requirements do *not* hold in the deep RL scenario. The challenges of extending RLSVI to deep RL are listed below.

1) **If the provided feature is not good and fixed, RLSVI is rarely competent.** Under the deep RL setting, only a raw feature outputted by a randomly initialized neural network is available. Such an unpolished feature has limited representation power, so the $Q$-value function cannot be approximated accurately. Further, the mechanism of RLSVI only involves updating the model parameter, while the feature is left to be fixed (as a raw feature in this setting). As a result, the empirical performance of RLSVI is often poor; see Appendix E.1 for the evidence.

2) **When we update the feature over iterations in deep RL, the computational complexity of RLSVI becomes unbearable.** As the data accumulates over iterations, RLSVI needs to repeatedly compute the feature covariance matrix to update the posterior distribution. With the fixed feature mapping, this process can be efficiently implemented in an incremental way (see (1.1), where $x_K$ denotes the state-action pair $(s_K, a_K)$ at iteration $K$ and $\phi : \mathcal{S} \times \mathcal{A} \to \mathbb{R}^d$ is a feature mapping.). However, when the feature mapping $\phi_K$ is changing over iterations, we need to repeatedly calculate the matrix $\Phi_K$ using all historical data as in (1.2) (e.g. in Atari, this calculation could involve more than 1M samples with dimension 512), and this process results in a huge computation burden.

$$\text{fixed } \phi: \quad \Phi_K = \Phi_{K-1} + \phi(x_K)\phi(x_K)^\top \quad \text{with } \Phi_0 = \mathbf{0}, \tag{1.1}$$

$$\text{changing } \phi_K: \quad \Phi_K := \sum_{\ell=1}^{K} \phi_K(x_\ell)\phi_K(x_\ell)^\top, \; \Phi_{K-1} := \sum_{\ell=1}^{K-1} \phi_{K-1}(x_\ell)\phi_{K-1}(x_\ell)^\top, \cdots \tag{1.2}$$

To address these two issues in deep RL, Bootstrapped DQN (BootDQN) (Osband et al., 2016a; 2018) takes a remarkable first step. To bypass the hurdle of issues 1 and 2, BootDQN simultaneously trains tens of ensembles (i.e., randomly initialized neural networks) and views them as approximate posterior samples of $Q$-value functions. However, with limited computation resources, BootDQN typically uses ensembles with a strictly limited number of elements. As such, the quality of the learned posterior distribution could be poor (Lu & Van Roy, 2017).

In this work, we present a principled approach named *HyperDQN* that addresses the above issues under the context of deep RL. Specifically, HyperDQN is built on two parametric models: in addition to a non-linear neural network (i.e., base model) that predicts $Q$-values, it employs a hypermodel (Dwaracherla et al., 2020) (i.e., meta model). This hypermodel maps a vector $z$ (from the Gaussian distribution) to a parameter instance of the $Q$-value function. As shown in Figure 1, this hypermodel aims to serve as a proxy for generating the posterior samples of $\theta_{\text{predict}}$. To achieve this goal, both models are jointly optimized by a specially designed temporal difference (TD) objective (see (4.2)).

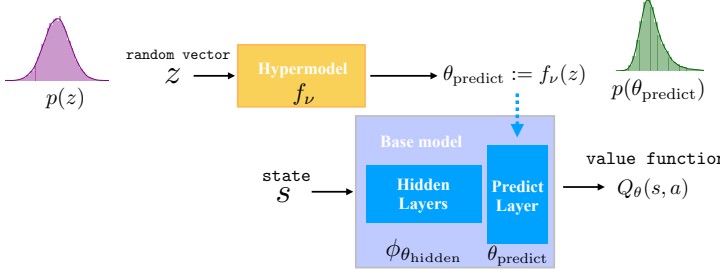

Figure 1: Illustration for HyperDQN .

After optimization, our approach has the following essential merits.

- First, we prove that the optimized hypermodel can approximate the posterior distribution under the Bayesian linear regression case (see Theorem 1). Thus, diverse $Q$-value functions can be sampled to select exploratory action sequences. This retains the punchline of RLSVI for efficient exploration. We remark that this theoretical guarantee is missing in (Dwaracherla et al., 2020).

- Second, the feature mapping $\phi_{\theta_{\text{hidden}}}(\cdot)$ is updated in an end-to-end way during the training. This deals with the feature learning problem of RLSVI (i.e., issue 1).

- Third, the posterior samples of $\theta_{\text{predict}}$ are obtained without involving (1.2). Specifically, the hypermodel could directly output an approximate posterior sample by taking $z$ as its input. This addresses the computational problem of RLSVI when the feature is changing (i.e., issue 2).

- Finally, compared with the *finite* ensembles in BootDQN, our approach learns the posterior distribution in a meta way and the hypermodel has certain generalization ability. Thus, our approach could have a better posterior approximation with the same computation resources. As such, our method is more computationally efficient to achieve a near-optimal performance.

To evaluate the efficiency of algorithms, we first consider the Atari suite (Bellemare et al., 2013) and assess the efficiency in terms of the human-normalized score over 49 tasks. The empirical result suggests HyperDQN with 20M frames outperforms DQN (Mnih et al., 2015) with 200M frames in terms of the maximum human-normalized score. With the same training frames, HyperDQN also outperforms the exploration bonus methods OPIQ (Rashid et al., 2020) and OB2I (Bai et al., 2021), and randomized exploration methods BootDQN (Osband et al., 2018) and NoisyNet (Fortunato et al., 2018). For another challenging benchmark SuperMarioBros (Kauten, 2018), HyperDQN beats these baselines on 5 out of 9 games in terms of the raw scores.

## 2 BACKGROUND

**Markov Decision Process.** In the standard reinforcement learning (RL) framework (Sutton & Barto, 2018), a learning agent interacts with an Markov Decision Process (MDP) to improve its performance via maximizing cumulative reward. The sequential decision process is characterized as follows: at each timestep $t$, the agent receives a state $s_t$ from the environment and selects an action $a_t$ from its policy $\pi(a|s) = \Pr\{a = a_t|s = s_t\}$; this decision is sent back to the environment, and the environment gives a reward signal $r(s_t, a_t)$ and transits to the next state $s_{t+1}$ based on the state transition probability $p^a_{s,s'} = \Pr\{s' = s_{t+1}|s = s_t, a = a_t\}$. The main target of RL is to maximize the (expected) discounted return $\mathbb{E}\big[\sum_{t=0}^{\infty} \gamma^t r(s_t, a_t)\big|s_0 \sim \rho(\cdot)\big]$, where $\rho(\cdot)$ is initial state distribution and $\gamma \in (0, 1)$ is a discount factor.

**Deep Q-Networks (DQN).** In Deep Q-Networks (DQN) (Mnih et al., 2015), it employs a neural network to approximate the $Q$-value function, which is defined as $Q^\pi(s, a) = \mathbb{E}[\sum_{t=0}^{\infty} \gamma^t r(s_t, a_t)|s_0 = s, a_0 = a]$. In particular, a *temporal difference* (TD) based objective is applied:

$$\min_\theta \sum_{(s,a,r,s')\sim\mathcal{D}} \big(r(s, a) + \gamma \max_{a'} Q_{\bar\theta}(s', a') - Q_\theta(s, a)\big)^2, \tag{2.1}$$

where $\mathcal{D}$ is the experience replay buffer, $Q_\theta$ is a prediction network parameterized by $\theta$, and $Q_{\bar\theta}$ is the so-called target network, which is a delayed copy of $Q_\theta$ for stable training.

## 3 RELATED WORK

**Epistemic uncertainty qualification.** It is justified that dithering strategies like $\epsilon$-greedy (Mnih et al., 2015) are inefficient (Osband et al., 2019). This is intuitive since they do not have any epistemic uncertainty measure and hence cannot "write-off" sub-optimal actions after experimentation. Importantly, epistemic uncertainty reflects the confidence about the unknown environment in online decision-making (Osband, 2016a). For tabular MDPs with the one-hot feature, "count" provides an epistemic uncertainty measure (Brafman & Tennenholtz, 2002; Jaksch et al., 2010; Azar et al., 2017; Jin et al., 2018) and "covariance" serves as the counterpart for linear MDPs (Jin et al., 2020; Cai et al., 2020). Currently, we do not have a perfect epistemic uncertainty qualification tool for deep RL, where a good feature is unknown in advance. As argued in (Osband et al., 2018), approaches like dropout (Srivastava et al., 2014) and distributional operator (Bellemare et al., 2017) are not suitable since they are designed for risk estimation. In this paper, we focus on the extension of hypermodel (Dwaracherla et al., 2020), which could capture the epistemic uncertainty for bandit tasks.

**OFU-based exploration.** Based on the mentioned uncertainty qualification, optimism in the face of uncertainty (OFU) based methods (Jaksch et al., 2010; Azar et al., 2017; Jin et al., 2018) construct upper confidence bound (UCB) to direct exploration. These theoretical works have inspired many empirical studies that encourage exploration by adding "reward/exploration bonus" to mimic UCB (Stadie et al., 2015; Pathak et al., 2017; Tang et al., 2017; Burda et al., 2019a;b). The challenges in this

direction include: how to obtain task-relevant reward bonus (O'Donoghue et al., 2018)? how to get an optimistic initialization with neural network (Rashid et al., 2020)? how to properly back-propagate the uncertainty over periods to induce temporally extended behaviors (Bai et al., 2021)? Though some of SOTA exploration bonus methods have achieved superior performance on *some* hard exploration tasks, (Taïga et al., 2020) report that these algorithms do not provide meaningful gains over the $\epsilon$-greedy scheme on the *whole* Atari suite.

**TS-based exploration.** On the other hand, Thompson sampling (TS) based methods design the exploration strategy in a Bayesian way (Osband et al., 2013; Osband & Van Roy, 2017). Randomized least-square value iteration (RLSVI) provides a promising direction (Osband et al., 2016b; 2019). As we have mentioned in Section 1, this method, however, can only be applied when a good feature is known and fixed during the training. Following RLSVI, Osband et al. (2016a) develop a practical algorithm called Bootstrapped DQN, which uses finite ensembles to generate the randomized value functions. As discussed earlier, the main bottleneck of Bootstrapped DQN is its computation complexity: it requires lots of ensembles to obtain an accurate approximation of the posterior samples (Lu & Van Roy, 2017). Our method is closely related to NoisyNet (Fortunato et al., 2018) as both methods inject noise to the parameter. However, we cannot view NoisyNet as an extension of RLSVI. The main reason is that NoisyNet is not ensured to approximate the posterior distribution as stated in (Fortunato et al., 2018). Besides, Ishfaq et al. (2021) attempt to extend the idea of RLSVI to the general case. In particular, their algorithm solves $M$ randomized least-square problems and chooses the most optimistic value function in each iteration. Since $M$ is selected to be proportional to the inherent dimension in theory, their algorithm is also not computationally efficient.

## 4 METHODOLOGY

### 4.1 ARCHITECTURE DESIGN

In this part, we explain the architecture design of HyperDQN shown in Figure 1. First, the hypermodel $f_\nu(\cdot)$ maps a $N_z$-dimension random vector $z \sim p(z)$ (e.g., a Gaussian distribution) to a specific $d$-dimension parameter $\theta := f_\nu(z)$. For example, if $z$ follows an isotropic Gaussian distribution $\mathcal{N}(0, I)$ and the hypermodel $f_\nu(z) = \nu_w^\top z + \nu_b$ is a linear model, then $\theta := f_\nu(z) \sim \mathcal{N}(\nu_b, \nu_w^\top \nu_w)$, where $\nu_w \in \mathbb{R}^{N_z \times d}$ and $\nu_b \in \mathbb{R}^d$ are the weight and bias of the linear hypermodel with $\nu = (\nu_w, \nu_b)$. In general, the hypermodel could be a non-linear mapping so that $f_\nu(z)$ could follow an arbitrary distribution. Unless stated otherwise, we only consider the linear hypermodel rather than non-linear ones. This is because linear hypermodel has sufficient representation power to approximate the posterior distribution (like RLSVI does) (Dwaracherla et al. (2020)).

Now, we explain the basemodel in Figure 1. The base model (i.e., a deep neural network) operates in a standard way: it takes a state $s$ as its input and outputs $Q_\theta(s, a) \in \mathbb{R}$ for some action $a$. Concretely, the base model employs a feature extractor $\phi_{\theta_{\text{hidden}}} : \mathcal{S} \to \mathbb{R}^d$ (i.e., the hidden layers in Figure 1) to learn a good representation. Then, the base model outputs $Q(s, a)$ by $Q(s, \cdot) = \theta_{\text{predict}}^\top \phi_{\theta_{\text{hidden}}}(s)$, where $\theta_{\text{predict}} \in \mathbb{R}^{d \times N_a}$ is the parameter of the prediction layer (i.e., the last layer of the base model in Figure 1). By our formulation, $\theta_{\text{predict}}$ is also the output of the hypermodel. That is, $\theta_{\text{predict}} = f_\nu(z) + f_{\nu_{\text{prior}}}(z)$, where $f_{\nu_{\text{prior}}}(z)$ is a fixed prior hypermodel; see the discussion below. Numerically, we fix $v_{\text{prior}} = v_0$, where $v_0$ is a random initialization. In the sequel, we will introduce the difference between the hypermodel in Figure 1 and that in Dwaracherla et al. (2020).

### 4.2 DIFFERENCES WITH DWARACHERLA ET AL. (2020): HOW THE DIRECT EXTENSION FAILS

Readers may notice that our architecture design is different from the original one in (Dwaracherla et al., 2020). Concretely, we apply the hypermodel for the *last* layer of the base model, but in (Dwaracherla et al., 2020) the hypermodel is applied for *all* layers of the base model. This modification is aimed to overcome the training difficulty: the training will fail if otherwise (see Appendix E.2 for the numerical evidence). Why does the training fail under the original design? We find out it is due to the severe gradient explosion at the initialization, which does not happen if no hypermodel is attached. For standard deep neural network models, such gradient explosion is mainly avoided by carefully designed initialization strategies (Sun, 2019). For instance, the famous LeCun initialization rule (LeCun et al., 1998) suggests that we should initialize the parameter of $i$-th layer by sampling from the Gaussian distribution $\mathcal{N}(0, \sigma^2)$ with $\sigma = 1/\sqrt{d_{i-1}}$, where $d_{i-1}$ is the width of $(i-1)$-th layer. As a result, the input signal and the output signal have the same order $\ell_2$-norm after processing.

However, things have changed if we directly use the architecture in (Dwaracherla et al., 2020) with this initialization technique: the output of the hypermodel is expected to lie between $-1$ and $1$ (up to constants); this further implies that the parameter is in $[-1, 1]$ for each layer of the base model, which disobeys the principle $1/\sqrt{d_{i-1}}$. Consequently, the input signal amplifies over layers and the gradient explodes. To bypass the training issue, Dwaracherla et al. (2020) use a shadow base model (e.g., 2 layers with the width of 10) and further implement the blocked hypermodel; see (Dwaracherla et al., 2020, Section 4) for details. But for deep RL, this training issue is severe and we overcome this challenge by the proposed architecture shown in Figure 1, in which the hidden layers of the base model are initialized with common techniques and the last layer could be properly initialized by normalizing the output of the hypermodel. This deals with the mentioned parameter initialization issue. Fortunately, this simple architecture still retains the main ingredient of RLSVI (i.e., capturing the posterior distribution over a linear prediction function).

## 4.3 TRAINING OBJECTIVE

In this part, we introduce the objective function. For the sake of better presentation, let us first consider a regression task. Let $x \in \mathbb{R}^d$ be the feature and $y \in \mathbb{R}$ be the label, the objective function developed in (Dwaracherla et al., 2020) for training the hypermodel is

$$L(\nu; \mathcal{D}) = \int_z p(z) \left[ \sum_{(x,y,\xi) \in \mathcal{D}} \left( y + \underbrace{\sigma_\omega z^\top \xi}_{(a)} - \underbrace{(g_{f_{\nu_{\text{prior}}}(z)}(x) + g_{f_\nu(z)}(x))}_{(b)} \right)^2 + \underbrace{\frac{\sigma_\omega^2}{\sigma_p^2} \|f_\nu(z)\|^2}_{(c)} \right] (\mathrm{d}z),$$

(4.1)

where $\xi$ is a random vector independently sampled from the unit hypersphere, paired with each $(x, y)$ in the dataset $\mathcal{D} = \{(x_i, y_i, \xi_i) : i = 1, \ldots, |\mathcal{D}|\}$; $\sigma_\omega > 0$ is the noise scale and $\sigma_p > 0$ is the regularization scale. Next, we briefly explain three key parts in (4.1). First, $(a)$ is an artificial noise term exerted on the label $y$, which is introduced for pure technical purpose. We defer more discussion of $(a)$ to Remark 1. Second, $(b)$ contains two base models denoted by $g$. Concretely, $g_{f_{\nu_{\text{prior}}}(z)}(\cdot)$ is a *prior* model parameterized by the output of a fixed hypermodel $f_{\nu_{\text{prior}}}$. This term stores the prior information. At the same time, $g_{f_\nu(z)}(\cdot)$ is a *differential* model linked by the output of a trainable hypermodel $f_\nu$. We remark that term $(b)$ is called "additive prior models" in (Osband et al., 2018; Dwaracherla et al., 2020); see (Dwaracherla et al., 2020, Section 2.5) for a detailed discussion. Finally, $(c)$ is a regularization term, which is an essential design in Bayesian learning.

To address the limitations of RLSVI for deep RL tasks, we first employ a feature mapping (parameterized by $\theta_{\text{hidden}}$) and approximate the $Q$-value function by a linear function (parameterized by $\theta_{\text{predict}}$) over the learned feature. Furthermore, the parameter of this linear function is modeled by the hypermodel (i.e., $\theta_{\text{predict}} = f_{\nu_{\text{prior}}}(z) + f_\nu(z)$) as discussed in Section 4.1. To this end, we extend (4.1) and implement the following optimization problem:

$$\min_{\nu, \theta_{\text{hidden}}} \int_z p(z) \left[ \sum_{(s,a,r,\xi,s') \in \mathcal{D}} \left( Q_{\text{target}}(s', z) + \sigma_\omega z^\top \xi - Q_{\text{prediction}}(s, a, z) \right)^2 + \frac{\sigma_\omega^2}{\sigma_p^2} \|f_\nu(z)\|^2 \right] (\mathrm{d}z),$$

(4.2)

where

$$Q_{\text{prediction}}(s, a, z) = Q_{\theta_{\text{prior}}, f_{\nu_{\text{prior}}}(z)}(s, a) + Q_{\theta_{\text{hidden}}, f_\nu(z)}(s, a),$$

$$Q_{\text{target}}(s', z) = r + \gamma \max_{a'} \left[ Q_{\theta_{\text{prior}}, f_{\nu_{\text{prior}}}(z)}(s', a') + Q_{\bar{\theta}_{\text{hidden}}, f_{\bar\nu}(z)}(s', a') \right].$$

(4.3)

Similar to the formulation in the regression problem, the $Q$-value function is the sum of the prior $Q_{\theta_{\text{prior}}, f_{\nu_{\text{prior}}}(z)}$ and the differential $Q_{\theta_{\text{hidden}}, f_\nu(z)}$. Following the target network in DQN, $(\bar{\theta}_{\text{hidden}}, \bar\nu)$ is the delayed copy of $(\theta_{\text{hidden}}, \nu)$. Compared with the TD loss in (2.1), there is an additional noise term $\sigma_\omega z^\top \xi$ for Bayesian learning and the prediction layer of the $Q$-value function is modeled as a probabilistic layer by the hypermodel. In experiments, we use the average of finite samples of $z$ to approximate (4.2); see the empirical loss function in (A.6) in Appendix. We choose Adam (Kingma & Ba, 2015) to optimize the empirical loss function.

## 4.4 THEORETICAL GUARANTEE

To understand our method, we explain why (4.1) is a good objective to generate approximate posterior samples. We provide the analysis under the linear case.

**Assumption 1.** *Suppose the data generation follows $y = x^\top \theta^\star + \omega^\star$, $\omega^\star \sim \mathcal{N}(0, \sigma_\omega^2)$ and the prior distribution over $\theta^\star$ is $\mathcal{N}(\bar{\theta}_p, \sigma_p^2 I)$. Furthermore, assume the base model is linear, i.e., $g_{f_\nu(z)}(x) = x^\top f_\nu(z)$ and $g_{f_{\nu_{prior}}(z)}(x) = x^\top f_{\nu_{prior}}(z)$. Moreover, assume the hypermodel is also linear, i.e., $f_\nu(z) = \nu_w^\top z + \nu_b$ and $f_{\nu_{prior}}(z) = \nu_w^{prior\top} z + \nu_b^{prior}$; $\nu = (\nu_w, \nu_b)$ and $\nu_{prior} = (\nu_w^{prior}, \nu_b^{prior})$.*

Based on Assumption 1, the posterior distribution over $\theta^\star$ is $\mathcal{N}(\mathbb{E}[\theta^\star \mid \mathcal{D}], \text{Cov}[\theta^\star \mid \mathcal{D}])$ with

$$\mathbb{E}[\theta^\star \mid \mathcal{D}] = \left( \frac{1}{\sigma_\omega^2} X^\top X + \frac{1}{\sigma_p^2} I \right)^{-1} \left( \frac{1}{\sigma_\omega^2} X^\top Y + \frac{1}{\sigma_p^2} \bar{\theta}_p \right),$$

$$\text{Cov}[\theta^\star \mid \mathcal{D}] = \left( \frac{1}{\sigma_\omega^2} X^\top X + \frac{1}{\sigma_p^2} I \right)^{-1},$$

where $X \in \mathbb{R}^{|\mathcal{D}| \times d}$ and $Y \in \mathbb{R}^{|\mathcal{D}|}$ are concatenation of $x_i$ and $y_i$, respectively. Also, we have that $g_{f_{\nu_{prior}}(z)}(x) + g_{f_\nu(z)}(x) = g_{f_{\nu_{prior}}(z) + f_\nu(z)}(x) = g_\theta(x) = x^\top \theta$, where $\theta := f_{\nu_{prior}}(z) + f_\nu(z)$.

**Theorem 1** (Informal). *Under Assumption 1, set $\nu_w^{prior} = \sigma_p I$ and $\nu_b^{prior} = \bar{\theta}_p$. Let $\nu^\star = (\nu_w^\star, \nu_b^\star)$ be the optimal solution of (4.1) conditioned on specific realizations of $\xi$, then $\theta := f_{\nu_{prior}}(z) + f_{\nu^\star}(z) \sim \mathcal{N}(\nu_b^{prior} + \nu_b^\star, (\nu_w^{prior} + \nu_w^\star)^\top (\nu_w^{prior} + \nu_w^\star))$ satisfies*

$$\nu_b^{prior} + \nu_b^\star = \mathbb{E}[\theta^\star \mid \mathcal{D}], \quad (\nu_w^{prior} + \nu_w^\star)^\top (\nu_w^{prior} + \nu_w^\star) = \text{Cov}[\theta^\star \mid \mathcal{D}] + \texttt{err}(\xi).$$

*Furthermore, the error term satisfies $\mathbb{E}_\xi[\texttt{err}(\xi)] = 0$.*

Theorem 1 states that the optimized hypermodel can approximate the true posterior distribution. The formal statement and proof are given in Appendix B.3. We emphasize that the message in Theorem 1 is *not* discussed in (Dwaracherla et al., 2020, Theorem 1). Dwaracherla et al. (2020) prove that a linear hypermodel has sufficient representation power to approximate any distribution (over functions) so it is unnecessary to use a non-linear one. However, this guarantee is unrelated to the objective (4.1): Dwaracherla et al. (2020) only show there *exists* a linear hypermodel can approximate the posterior distribution but Dwaracherla et al. (2020) do not tell us whether (4.1) can lead to such a hypermodel. Instead, this question is affirmatively answered by Theorem 1. In addition, Theorem 1 also conveys an important message about the noise used in (4.1), which we explain in Remark 1.

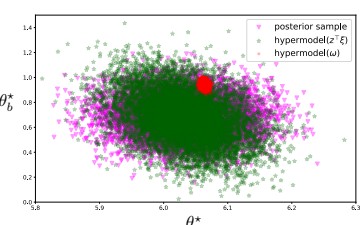

Figure 2: Visualization of true posterior samples and learned posterior samples.

**Remark 1.** *Readers may ask whether an independent Gaussian noise $\omega$ (rather than the $z$-dependent noise $z^\top \xi$) is applicable in (4.1). Intuitively, $\omega$ can also form a randomized-least square problem to optimize the hypermodel. However, by inspecting the proof of Theorem 1, we claim that this scheme does not work. Technically speaking, $\omega$ will introduce exogenous random sources into the objective function, which jeopardizes the learning goal of mapping an index $z$ to a posterior sample $\theta$; see Appendix B.4 for a formal argument.*

*To better understand the fundamental issue involved here, we run simulation on a toy 2-dimensional Bayesian linear regression problem (Assumption 1) with $\theta^\star = (\theta_w^\star, \theta_b^\star)$; see Appendix C.3 for experiment details. After optimizing (4.1), we visualize the posterior samples generated by the linear hypermodel in Figure 2. In particular, $\texttt{hypermodel}(z^\top \xi)$ (in green) and $\texttt{hypermodel}(\omega)$ (in red) correspond to learned samples via solving (4.1) with a $z$-dependent noise $z^\top \xi$ and an independent Gaussian noise $\omega$, respectively. The variant $\texttt{posterior sample}$ (in purple) is obtain by the true posterior distribution $\mathcal{N}(\mathbb{E}[\theta^\star \mid \mathcal{D}], \text{Cov}[\theta^\star \mid \mathcal{D}])$. We observe that the $z$-dependent noise is indispensable for the hypermodel to approximate the posterior distribution.*

### 4.5 THE ALGORITHM

We outline the proposed method in Algorithm 2 in Appendix. At the beginning of each episode, a random vector $z$ is sampled to obtain the $Q$-value function, which is further used for interactions with environments. Subsequently, the randomized temporal difference objective in (4.2) is optimized to train the hypermodel and the hidden layers based on the collected data.

Next, we interpret HyperDQN from an algorithmic level introduced in (Osband et al., 2018). In particular, HyperDQN achieves the desiderata:

- **Task-relevant feature.** A task-specific feature can be obtained by the end-to-end training in (4.2). Such a feature is good in the sense that it is optimized to approximate multiple value functions as different invocations of $z$ yield several individual TD loss functions. As discussed previously, this feature learning procedure is beyond the scope of RLSVI.

- **Commitment.**[2] The agent executes action sequences that span multiple periods by obeying its intent. In each episode, HyperDQN samples a specific value function and takes greedy actions according to this value function until the episode terminates. Unlike BootDQN (Osband et al., 2016a) [3] and OFU-based algorithms (Taïga et al., 2020; Rashid et al., 2020; Bai et al., 2021), HyperDQN does *not* combine $\epsilon$-greedy as the perturbation in $\epsilon$-greedy ruins "far-sighted" behaviors generated by the original algorithm.

- **Cheap computation.** As discussed earlier, RLSVI needs to re-compute the feature covariance matrix when the feature mapping changes, which results in a huge computation burden. To address this issue, BootDQN simultaneously trains tens of ensembles. When the computation resources are limited, the quality of generated posterior samples by BootDQN is poor (Lu & Van Roy, 2017). However, our approach learns the approximate posterior samples in a meta way and the hypermodel has certain generalization ability. As such, and our method tends to be more computation efficient.

As a side note, if the random index $z$ is repeatedly sampled from a finite set, the hypermodel degenerates to finite ensembles. As such, BootDQN can be viewed as a special case of HyperDQN. Following (Osband et al., 2018), we summarize BootDQN and HyperDQN in Table 1.

Table 1: Important issues in posterior approximations for deep RL. A green tick indicates a satisfying result, a red cross implies an undesirable result and a yellow circle means something in between.

| | Task-relevant feature | Commitment | Cheap computation |
|---|:---:|:---:|:---:|
| BootDQN (Osband et al., 2018) | ✓ | 🟡 | ✗ |
| **HyperDQN** | ✓ | ✓ | 🟡 |
| RLSVI (Osband et al., 2016b) | ✗ | ✓ | ✗✗ |

## 5 EXPERIMENTS

In this section, we present numerical experiments to validate the efficiency of the proposed method. Experiment details can be found in Appendix C.

### 5.1 ATARI

Our first experiment is on the Arcade Learning Environment (Bellemare et al., 2013), which provides a platform to assess the general competence. The training and evaluation procedures (e.g., observation preprocessing and reward clipping) follows (Mnih et al., 2015) and (van Hasselt et al., 2016). We measure the sample efficiency in terms of the human-normalized score during the interaction. In particular, the raw score of each game is normalized so that $0\%$ corresponds to a random agent and $100\%$ to a human expert.

We consider five baselines: DQN (Mnih et al., 2015), OPIQ[4] (Rashid et al., 2020), OB2I[5] (Bai et al., 2021), BootDQN[6] (Osband et al., 2018) and NoisyNet[7] (Fortunato et al., 2018). In particular, OPIQ and OB2I are two advanced OFU (i.e, exploration bonus based) methods while BootDQN and NoisyNet are randomized exploration methods.

The learning curve over 49 games (by median) with the 20M frames training budget is displayed in Figure 3. We see that HyperDQN quickly explores and achieves the best performance among baselines. The performance of OFU-based methods is bad: OPIQ cannot achieve a satisfying

---

[2]This concept is originally defined in the context of concurrent RL (Dimakopoulou & Roy, 2018). We borrow this concept to help discuss the combination with $\epsilon$-greedy under the standard RL framework.

[3]BootDQN uses $\epsilon$-greedy for complicated tasks like Atari; see (Osband et al., 2016a, Appendix D.1).

[4]https://github.com/oxwhirl/opiq

[5]https://github.com/Baichenjia/OB2I

[6]Modified from https://github.com/johannah/bootstrap_dqn for a fair comparison.

[7]Modified from https://github.com/ku2482/fqf-iqn-qrdqn.pytorch/blob/master/fqf_iqn_qrdqn/network.py for a fair comparison.

Table 2: Comparison of algorithms on Atari in terms of the median over 49 games' maximum human-normalized scores. Note that the performance of DQN is based on 200M training frames while other methods are based on 20M training frames.

| DQN (**200M**) | OPIQ | OB2I | BootDQN | NoisyNet | HyperDQN |
|---|---|---|---|---|---|
| 93% | 37% | 50% | 82% | 91% | **110%** |

performance over all games and OB2I degenerates after 10M frames. This is partially because the exploration bonus may guide a direction that is unrelated to the environment reward. However, randomized exploration methods do not have such an issue. Our experiment results regarding OFU-based methods are consistent with (Taïga et al., 2020): though they may perform well on some hard exploration tasks, current OFU-based methods cannot provide meaningful gains over the vanilla DQN method on the whole Atari suite.

We notice that researchers also consider the metric of the maximum human-normalized score, which measures the performance of the best policy during training; refer to (Mnih et al., 2015; van Hasselt et al., 2016; Hessel et al., 2018; Fortunato et al., 2018). Specifically, we compute the maximum evaluation scores after training for each game and average them by the median. We report such results in Table 2. In particular, we see that HyperDQN outperforms the performance obtained by DQN with 200 training frames. We do not consider the statistics of mean because it is significantly affected by some special games (e.g., Atlantis); see Table 5 in Appendix.

We note that the 20M frames training budget (rather than 200M frames) is commonly used in (Lee et al., 2019; Bai et al., 2021). The main reason is that algorithms can achieve satisfactory performance on many tasks with 20M frames while the 200M frames training budget requires about 30/10 days with a CPU/GPU machine for a game, which is expansive in time and money.

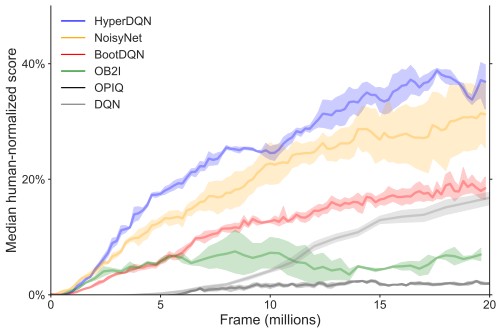
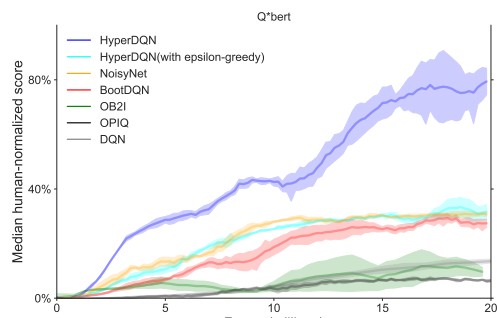

Figure 3: Comparison of algorithms in terms of the human-normalized score over 49 games in Atari.

Figure 4: Comparison of algorithms on Q*bert. $\epsilon$-greedy makes HyperDQN worse.

To better understand the performance of HyperDQN, we visualize the relative improvements over baselines on "easy exploration" and "hard exploration" environments in Figure 12, 13, 14, and 15 in Appendix D. We see that HyperDQN could perform well on many easy and hard exploration tasks. However, HyperDQN does not work for Montezuma's Revenge, in which the reward is very sparse so there is limited feedback for feature selection. For the same reason, randomized exploration methods in (Osband et al., 2018; Fortunato et al., 2018) do not perform well on this task.

Finally, we provide evidence that commitment is crucial for our method. Specifically, $\epsilon$-greedy would lead to worse performance for HyperDQN; see the empirical result on the game Q*bert in Figure 4. The same observation holds for other environments (refer to Appendix D).

## 5.2 SUPERMARIOBROS

In this part, we evaluate algorithms on the SuperMarioBros suite (Kauten, 2018). Environment preprocessing and algorithm parameters basically follow the one used in Atari and we do not tune parameters for any algorithm.

We note that SuperMarioBros-1-3 and SuperMarioBros-2-2 are two hard exploration tasks due to the long planning horizon and sparse reward. Experiments are run with 3 random seeds. Similar to Table 5, we report the maximum scores in the following Table 3; see Figure 17 in Appendix for the learning curves on each game. We see that HyperDQN beats baselines on 5 out of 9 games.

Table 3: Comparison of algorithms on SuperMarioBros in terms of the raw scores by the best policies with 20M training frames.

| | DQN | OPIQ | OB2I | BootDQN | NoisyNet | HyperDQN |
|---|---|---|---|---|---|---|
| SuperMarioBros-1-1 | $1,070$ | $7,650$ | $4,457$ | $7,009$ | $\mathbf{12,439}$ | $7,924$ |
| SuperMarioBros-1-2 | $2,883$ | $5,515$ | $4,695$ | $5,665$ | $6,347$ | $\mathbf{8,267}$ |
| SuperMarioBros-1-3 | $667$ | $2,053$ | $1,583$ | $1,609$ | $1,587$ | $\mathbf{6,047}$ |
| SuperMarioBros-2-1 | $10,800$ | $21,654$ | $14,226$ | $\mathbf{26,415}$ | $14,017$ | $23,047$ |
| SuperMarioBros-2-2 | $813$ | $1,630$ | $1,588$ | $1,092$ | $1,808$ | $\mathbf{1,984}$ |
| SuperMarioBros-2-3 | $3,373$ | $4,718$ | $4,402$ | $5,108$ | $\mathbf{6,490}$ | $5,980$ |
| SuperMarioBros-3-1 | $2,560$ | $3,700$ | $3,251$ | $3,862$ | $11,310$ | $\mathbf{48,385}$ |
| SuperMarioBros-3-2 | $11,633$ | $20,872$ | $26,508$ | $20,955$ | $33,489$ | $\mathbf{41,140}$ |
| SuperMarioBros-3-3 | $1,007$ | $2,440$ | $3,009$ | $2,650$ | $\mathbf{5,886}$ | $5,568$ |

## 5.3 DEEP SEA

In this part, we consider the standard testbed for hard exploration: deep sea (Osband et al., 2018; 2020). In particular, there are two actions in this environment: move left and move right; see Figure 5. The reward is sparse and is only released when the agent always takes the "right" action to obtain the treasure at the corner. The maximum episode return is 0.99.

Following (Dwaracherla et al., 2020), we measure the computation complexity by

$$\text{computation complexity} = n_{\text{sgd}} \times n_z \times K, \tag{5.1}$$

where $n_{\text{sgd}}$ is the number of SGD steps per iteration, $n_z$ is the number of ensemble (index) samples, and $K$ is the minimum number of episode with return 0.99. This criterion is reasonable since without the restriction of the computation complexity, methods may perfectly approximate RLSVI and enjoy the same sample complexity.

The averaged empirical result with 10 random seeds is shown in Figure 6. In particular, the $x$-axis corresponds to the size of the deep sea, i.e., the number of rows in Figure 5, and the $y$-axis corresponds to the computation complexity as in (5.1). We see that HyperDQN is more computationally efficient than BootDQN. Other methods fail to solve the deep sea when the size is larger than 20, which is also observed in (Osband et al., 2018).

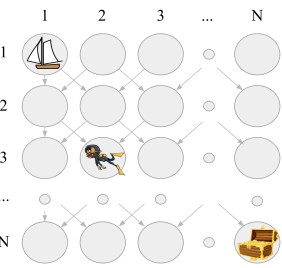

Figure 5: Illustration for deep sea.

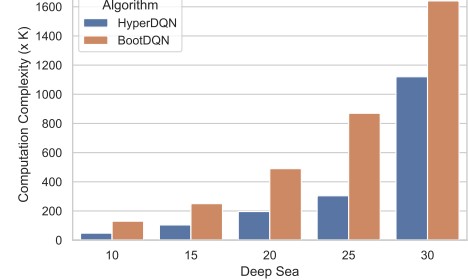

Figure 6: Comparison of HyperDQN and BootDQN in terms of computation complexity on the deep sea.

## 6 CONCLUSION AND FUTURE WORK

In this work, we present a practical exploration method to address the limitations of RLSVI and BootDQN. To reinforce the central idea, we leverage the hypermodel (Dwaracherla et al., 2020) and extend it from the bandit tasks to RL problems. Several algorithmic designs are developed to release the power of randomized exploration.

Future directions include extending the idea for continuous control (Lillicrap et al., 2016; Haarnoja et al., 2018) (i.e., building a randomized actor-critic with a similar architecture with HyperDQN; see Appendix E.5), utilizing the developed epistemic uncertainty qualification tool for offline RL (Fujimoto et al., 2019; Levine et al., 2020), and acquiring an informative prior from human demonstrations to accelerate exploration (Hester et al., 2018; Sun et al., 2018) (see Appendix E.6).

## ACKNOWLEDGMENTS AND DISCLOSURE OF FUNDING

We thank Chenjia Bai for sharing some training results of OB2I, Hao Liang for the insightful discussion, and Tian Xu for reading the manuscript and providing valuable comments. The work of Z.-Q. Luo is supported by the National Natural Science Foundation of China (No. 61731018) and the Guangdong Provincial Key Laboratory of Big Data Computation Theories and Methods.

## ETHICS STATEMENT

This work focuses on designing an efficient exploration method to improve the sample efficiency of reinforcement learning. This work may help reinforcement learning to be better used in the real world. There could be some unexpected consequences if the reinforcement learning is abused. For example, a robot is trained maliciously to hurt people.

## REPRODUCIBILITY STATEMENT

First, a formal statement and proof of Theorem 1 are given in Appendix B.3. Second, the implementation details of the proposed algorithm HyperDQN can be found in Appendix A.2. Third, the experiment details of numerical results in Section 5 are provided in Appendix C.

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

# APPENDIX: HYPERDQN: A RANDOMIZED EXPLORATION METHOD FOR DEEP REINFORCEMENT LEARNING

## CONTENTS

# A  ALGORITHM DETAILS

## A.1  RANDOMIZED LEAST-SQUARE VALUE ITERATION (RLSVI)

In this part, we offer a brief introduction to randomized least-square value iteration (RLSVI) (Osband et al., 2016b; 2019). We hope this helps readers understand how RLSVI works and the underlying limitations of RLSVI.

In its original form, RLSVI is designed for the episodic Markov Decision Process for easy analysis. Here the word of "episodic" means the planning horizon (i.e., the episode length) is a finite number $T > 0$. Given a feature map $\phi : \mathcal{S} \times \mathcal{A} \to \mathbb{R}^d$ in advance, RLSVI assumes the optimal $Q$-value function is a linear model with respect to this feature map. That is, for any stage $t = 0, 1, \cdots, T - 1$, RLSVI assumes

$$Q_t^\star(s, a) = \phi_t(s, a)^\top \theta_t^\star, \tag{A.1}$$

where $\theta_t^\star \in \mathbb{R}^d$ is the unknown optimal parameter and $Q_t^\star$ is the optimal value function at stage $t$. Note that $\phi_t$ differs from stage to stage, but they are all fixed along the training.

With the above assumption, RLSVI takes three steps to balance the trade-off between exploration and exploitation.

- **(Updating)** Conditioned on the collected data, RLSVI estimates the optimal parameter $\theta_t^\star$ in a Bayesian way. In particular, RLSVI obtains the posterior distribution over $\theta_t^\star$ via solving a Bayesian linear regression problem[8]. More specifically, the Bayesian linear regression problem consists of the fixed feature $\phi_t$ and the label $y_t$ generated by dynamic programming; see (A.3). Furthermore, the posterior distribution can be obtained by the closed-form solution in (A.4).

- **(Sampling)** With the posterior distribution over $\theta_t^\star$, RLSVI samples a specific parameter $\widetilde{\theta}_t$, which forms the $Q$-value function $\widetilde{Q}_t(s, a) = \phi_t(s, a)^\top \widetilde{\theta}_t$.

- **(Interaction)** RLSVI takes the greedy action $\mathrm{argmax}_a \widetilde{Q}(s_t, a)$ at stage $t$ to collect new data.

Let $k$ indicates the episode count and $t$ indicates the stage/period count. Let $[T]$ indicates the set $\{0, 1, 2, \cdots, T - 1\}$. For instance, $\theta_{k,t}^\dagger \in \mathbb{R}^d$ means the posterior mean at episode $k$ and stage $t$. With these notations, the procedure of RLSVI is outlined in Algorithm 1.

---

**Algorithm 1** RLSVI (Osband et al., 2016b; 2019)

---

1: posterior mean $\theta_{0,t}^\dagger \leftarrow$ prior mean $\bar{\theta}_t$, posterior covariance $\Sigma_{0,t}^\dagger \leftarrow$ prior covariance $\sigma_p^2 I, \forall t \in [T]$.
2: sample $\widetilde{\theta}_t \sim \mathcal{N}(\theta_{0,t}^\dagger, \Sigma_{0,t})$ for $t \in [T]$.
3: **for** episode $k = 0, 1, 2, \cdots$ **do**
4:     **for** stage $t = 0, 1, 2, \cdots, T - 1$ **do**                         ▷ Interaction
5:         observe state $s_t$.
6:         take the greedy action $a_t = \mathrm{argmax}_a \phi(s_t, a)^\top \widetilde{\theta}_t$.
7:         receive reward $r(s_t, a_t)$.
8:     **end for**
9:     **for** stage $t = T - 1, T - 2, \cdots, 0$ **do**                        ▷ Update
10:         $\theta_{k+1,t}^\dagger, \Sigma_{k+1,t}^\dagger \leftarrow$ update the posterior distribution (see (A.4)).
11:         sample $\widetilde{\theta}_t \sim \mathcal{N}(\theta_{k,t}^\dagger, \Sigma_{k,t}^\dagger)$.                        ▷ Sampling
12:     **end for**
13: **end for**

---

For Line 10 in Algorithm 1, RLSVI leverages the tool of Bayesian linear regression. Abstractly, given the feature matrix $X \in \mathbb{R}^{N \times d}$ ($N$ is the number of samples) and the label vector $Y \in \mathbb{R}^N$, the posterior distribution over $\theta^\star$ can be computed in a closed-form way. More concretely, RLSVI forms

---

[8]Bayesian linear regression is reviewed in Appendix B.1.

$X_t$ and $Y_t$ with the state-action pairs collected at stage $t$ up to episode $k$:

$$X_t = \begin{bmatrix} \phi_t(s_{0,t}, a_{0,t}) \\ \phi_t(s_{1,t}, a_{1,t}) \\ \cdots \\ \phi_t(s_{k,t}, a_{k,t}) \end{bmatrix} \in \mathbb{R}^{(k+1) \times d}, \quad Y_t = \begin{bmatrix} y_{0,t} \\ y_{1,t} \\ \cdots \\ y_{k,t} \end{bmatrix} \in \mathbb{R}^{k+1}, \tag{A.2}$$

where

$$y_{k,t} = \begin{cases} r_{k,t} + \max_a (\phi_{t+1}^\top \widetilde{\theta}_{t+1})(s_{k,t+1}, a) & \text{if } t < T - 1 \\ r_{k,t} & \text{if } t = T - 1 \end{cases} \tag{A.3}$$

With such defined $X_t$ and $Y_t$, RLSVI can compute the posterior distribution $\mathcal{N}(\mathbb{E}[\theta_t^\star \mid \mathcal{D}], \mathrm{Cov}[\theta_t^\star \mid \mathcal{D}])$ over $\theta_t^\star$:

$$\mathbb{E}[\theta_t^\star \mid \mathcal{D}] := \theta_{k+1,t}^\dagger = \left( \frac{1}{\sigma_\omega^2} X_t^\top X_t + \frac{1}{\sigma_p^2} I \right)^{-1} \left( \frac{1}{\sigma_\omega^2} X_t^\top Y_t + \frac{1}{\sigma_p^2} \bar{\theta}_t \right),$$

$$\mathrm{Cov}[\theta_t^\star \mid \mathcal{D}] := \Sigma_{k+1,t}^\dagger = \left( \frac{1}{\sigma_\omega^2} X_t^\top X_t + \frac{1}{\sigma_p^2} I \right)^{-1}. \tag{A.4}$$

**Remark 2.** As the feature is fixed in RLSVI, there exists an efficient implementation of RLSVI based on the incremental update. Specifically, let $\Phi$ be the feature covariance matrix, i.e.,

$$\Phi_{k+1,t} = \frac{1}{\sigma_\omega^2} X_t^\top X_t + \frac{1}{\sigma_p^2} I.$$

For easy presentation, let $\phi_{k,t}$ denote $\phi(s_{k,t}, a_{k,t})$. Then, we have the incremental update formula:

$$\Phi_{k+1,t} = \Phi_{k,t} + \frac{1}{\sigma_\omega^2} \phi_{k,t} \phi_{k,t}^\top, \quad \text{with } \Phi_{0,t} = \frac{1}{\sigma_p^2} I.$$

Furthermore, we have a more efficient update rule for the posterior covariance in (A.4) by the Sherman–Morrison formula (see Lemma 1). More concretely, we have that

$$\Sigma_{k+1,t}^\dagger := \Sigma_{k,t}^\dagger - \frac{(1/\sigma_\omega^2) \cdot \Sigma_{k,t}^\dagger \phi_{k,t} \phi_{k,t}^\top \Sigma_{k,t}^\dagger}{1 + (1/\sigma_\omega^2) \cdot \phi_{k,t}^\top \Sigma_{k,t}^\dagger \phi_{k,t}}, \quad \text{with } \Sigma_{0,t}^\dagger = \sigma_p^2 I.$$

As a side note, the term $X_t^\top Y_t$ in (A.4) can also be implemented in an incremental update way. In short, the posterior distribution is easy to compute when the feature is fixed.

**Lemma 1** (Sherman–Morrison formula). Suppose $A \in \mathbb{R}^{n \times n}$ is an invertible square matrix and $u, v \in \mathbb{R}^n$ are column vectors. Then $A + uv^\top$ is invertible if and only if $1 + v^\top A^{-1} u \neq 0$. In this case,

$$\left( A + uv^\top \right)^{-1} = A^{-1} - \frac{A^{-1} uv^\top A^{-1}}{1 + v^\top A^{-1} u}.$$

Here, $uv^\top$ is the outer product of two vectors $u$ and $v$.

## A.2 IMPLEMENTATION OF HYPERDQN

In this part, we provide more implementation details of HyperDQN.

Recall that we work with two value functions: the prior one and the differential one. The prior value function and the differential value function are independently constructed, meaning there are no shared parameters. The actual $Q$-value is the weighted sum of two terms:

$$Q(s, a, z) = \beta_{\text{prior}} \cdot Q_{\theta_{\text{prior}}, f_{\nu_{\text{prior}}}(z)}(s, a) + \beta_{\text{differential}} \cdot Q_{\theta_{\text{hidden}}, f_\nu(z)}(s, a), \tag{A.5}$$

where $\beta_{\text{prior}} > 0$ and $\beta_{\text{differential}} > 0$ are two positive scalars. When $\beta_{\text{prior}} = \beta_{\text{differential}} = 1$, we recover the formulation in (4.2).

The feature extractor (i.e., the hidden layers) are task-specific, which will be discussed in Appendix C. For the hypermodel, the differential part is initialized based on the default initializer of PyTorch while the initialization of the prior hypermodel is a little tricky. Following (Dwaracherla et al., 2020), each row of $\nu_w$ is sampled from the unit hypersphere and $\nu_b$ is based on the mean of the desired prior distribution. The purpose of this initialization is to guarantee the $\theta_{\text{prior}} = f_{\nu_{\text{prior}}}(z)$ follows a desired Gaussian distribution. For example, if we want to obtain a prior distribution $\mathcal{N}(\mathbf{1}, I)$ for

---

**Algorithm 2** HyperDQN

---

1: agent step $n \leftarrow 0$, train step $\ell \leftarrow 0$.
2: **for** episode $k = 0, 1, 2, \cdots$ **do**
3:     generate a random vector $z \sim \mathcal{N}(0, I)$.                           ▷ Sampling
4:     instantiate the $Q$-value function $Q_\theta(s, a, z)$.
5:     **for** stage $t = 0, 1, 2, \cdots, T - 1$ **do**                     ▷ Interaction
6:         observe state $s_t$.
7:         take the greedy action $a \leftarrow \mathrm{argmax}_a Q_\theta(s_t, a, z)$.
8:         receive the next state $s_{t+1}$ and reward $r(s_t, a_t)$.
9:         sample $\xi$ uniformly from unit hypersphere.
10:        store $(s_t, a_t, r, \xi, s_{t+1})$ into the replay buffer $\mathcal{D}$.
11:        agent step $n \leftarrow n + 1$.
12:        **if** `mod` (agent step $n$, train frequency $M$) $== 0$ **then**       ▷ Update
13:            sample a mini-batch $\widetilde{\mathcal{D}}$ of $(s, a, r, \xi, s')$ from the replay buffer $\mathcal{D}$.
14:            sample $N$ random vectors $z_i'$: $\widetilde{\mathcal{Z}} = \{z_i'\}_{i=1}^N$.
15:            optimize $\nu$ and $\theta_{\mathrm{hidden}}$ using the empirical loss function (A.6) with $\widetilde{\mathcal{D}}$ and $\widetilde{\mathcal{Z}}$.
16:            train step $\ell \leftarrow \ell + 1$.
17:        **end if**
18:        **if** `mod` (train step $\ell$, target update frequency $G$) $== 0$ **then**
19:            update the target network.
20:        **end if**
21:     **end for**
22: **end for**

---

$\theta_{\mathrm{prior}} := f_{\nu_{\mathrm{prior}}}(z)$, it suffices to set $\nu_b = \mathbf{1}$ and to uniformly sample $\nu_w$ from the unit hypersphere, where $\mathbf{1}$ is a vector with elements of $1$.

According to the *population* objective function in (4.2), the *empirical* objective function $\mathcal{L}(\nu, \theta_{\mathrm{hidden}}; \widetilde{\mathcal{Z}}, \widetilde{\mathcal{D}})$ given the set $\widetilde{\mathcal{Z}}$ and mini-batch $\widetilde{\mathcal{D}}$ is

$$\frac{1}{|\widetilde{\mathcal{Z}}|} \left( \sum_{z \in \widetilde{\mathcal{Z}}} \frac{|\mathcal{D}|}{|\widetilde{\mathcal{D}}|} \sum_{(s,a,r,\xi,s') \in \widetilde{\mathcal{D}}} \left( Q_{\mathrm{target}}(s', z) + \sigma_\omega z^\top \xi - Q_{\mathrm{prediction}}(s, a, z) \right)^2 + \frac{\sigma_\omega^2}{\sigma_p^2} \| f_\nu(z) \|^2 \right),$$
(A.6)

where

$$Q_{\mathrm{prediction}}(s, a, z) = \beta_{\mathrm{prior}} \cdot Q_{\theta_{\mathrm{prior}}, f_{\nu_{\mathrm{prior}}}(z)}(s, a) + \beta_{\mathrm{differential}} \cdot Q_{\theta_{\mathrm{hidden}}, f_\nu(z)}(s, a),$$

$$Q_{\mathrm{target}}(s', z) = r + \gamma \max_{a'} \left[ \beta_{\mathrm{prior}} \cdot Q_{\theta_{\mathrm{prior}}, f_{\nu_{\mathrm{prior}}}(z)}(s', a') + \beta_{\mathrm{differential}} \cdot Q_{\bar{\theta}_{\mathrm{hidden}}, f_{\bar{\nu}}(z)}(s', a') \right].$$

In practice, (A.6) can be efficiently solved by Adam (Kingma & Ba, 2015). In particular, we sample multiple $z$ and the mini-batch from the experience replay buffer to construct the empirical objective function. As such, Adam can be applied to optimize the hidden layers and the hypermodel as illustrated in Figure 1.

**Remark 3** (Feature Representation for Prior Value Functions). Different from supervised learning, the prior value function is crucial for online decision making (Osband et al., 2018). For the prior value functions, we not only need to consider the prior hypermodel but also the prior feature representation. Importantly, the feature extractor for the prior value function should be fixed rather than being shared with the differential part. In other words, we use two independent feature networks (i.e., the hidden layers) for the prior value function and the differential value function, respectively. Instead, the output of the prior value function would change if the prior value function and the differential value function share the same hidden layers, which violates the intuition that the prior value function should not be affected by the learning process.

### A.3   SAMPLING FROM UNIT HYPERSPHERE

For completeness, we describe the method for uniform sampling over unit hypersphere. There are two steps:

(1) Generate $\mathbf{x} = (x_1, x_2, \ldots, x_d)$, using a zero mean and unit variance Gaussian distribution. Thus, the probability density of $\mathbf{x}$

$$p(\mathbf{x}) = \frac{1}{(2\pi)^{\frac{d}{2}}} \exp\left(-\frac{x_1^2 + x_2^2 + \cdots + x_d^2}{2}\right)$$

is spherically symmetric.

(2) Normalize the vector $\mathbf{x} = (x_1, x_2, \ldots, x_d)$ to a unit vector, namely $\mathbf{x}/\|\mathbf{x}\|$, which gives a sample uniformly over the unit hypersphere. Note that once the vector is normalized, its coordinates are no longer statistically independent.

# B    CONNECTION BETWEEN BAYESIAN LINEAR REGRESSION AND HYPERMODEL

## B.1    BAYESIAN LINEAR REGRESSION

In this part, we review the Bayesian linear regression. Let $x \in \mathbb{R}^d$ be the feature and $y = \langle \theta^\star, x \rangle + \omega^\star$ be the label, where $\omega^\star \in \mathcal{N}(0, \sigma_\omega^2)$ is the observation noise. Here, we treat $\theta^\star$ as a random variable and pose a prior distribution $p_0 : \mathcal{N}(\bar{\theta}_p, \sigma_p^2 I)$ on $\theta^\star$. Given the dataset $\mathcal{D} = \{(x_i, y_i)\}_{i=1}^N$, we can update the posterior probability density over $\theta^\star$ by the Bayes rule:

$$p(\theta^\star \mid \mathcal{D}) \propto p\left(\{(x_i, y_i)\}_{i=1}^N \mid \theta^\star\right) \cdot p_0(\theta^\star).$$

Thanks to the conjugate prior, the posterior distribution over $\theta^\star$ is also a Gaussian distribution $\mathcal{N}(\mathbb{E}[\theta^\star \mid \mathcal{D}], \mathrm{Cov}[\theta^\star \mid \mathcal{D}])$ with

$$\mathbb{E}[\theta^\star \mid \mathcal{D}] = \left(\frac{1}{\sigma_\omega^2} X^\top X + \frac{1}{\sigma_p^2} I\right)^{-1} \left(\frac{1}{\sigma_\omega^2} X^\top Y + \frac{1}{\sigma_p^2} \bar{\theta}_p\right), \tag{B.1}$$

$$\mathrm{Cov}[\theta^\star \mid \mathcal{D}] = \left(\frac{1}{\sigma_\omega^2} X^\top X + \frac{1}{\sigma_p^2} I\right)^{-1}, \tag{B.2}$$

where

$$X = \begin{bmatrix} x_1^\top \\ x_2^\top \\ \cdots \\ x_N^\top \end{bmatrix} \in \mathbb{R}^{N \times d}, \quad Y = \begin{bmatrix} y_1 \\ y_2 \\ \cdots \\ y_N \end{bmatrix} \in \mathbb{R}^N.$$

In summary, given the collected dataset (i.e., the features and labels), we can obtain posterior samples through *sampling* from $\mathcal{N}(\mathbb{E}[\theta^\star \mid \mathcal{D}], \mathrm{Cov}[\theta^\star \mid \mathcal{D}])$.

Recently, Osband et al. (2019) provide another way of obtaining posterior samples through *optimizing* a randomized least square problem.

**Lemma 2** ((Osband et al., 2019)). *Let $x$ be the feature vector and the target $y$ be generated by $y = \langle \theta^\star, x \rangle + \omega^\star$, where $\omega^\star \sim \mathcal{N}(0, \sigma_\omega^2)$ is the noise and the prior on $\theta^\star$ is $\mathcal{N}(\bar{\theta}_p, \sigma_p^2 I)$. Let $\omega$ be the algorithmic noise that follows the same distribution with $\omega^\star$ (i.e., $\omega \sim \mathcal{N}(0, \sigma_\omega^2)$) and $\widehat{\theta} \sim \mathcal{N}(\bar{\theta}_p, \sigma_p^2 I)$ be a prior sample. Then,*

$$\underset{\theta}{\arg\min} \sum_{(x,y,\omega) \in \mathcal{D}} (y + \omega - \theta^\top x)^2 + \frac{\sigma_\omega^2}{\sigma_p^2} \left\|\theta - \widehat{\theta}\right\|_2^2, \tag{B.3}$$

*or*

$$\widehat{\theta} + \underset{\theta}{\arg\min} \sum_{(x,y,\omega) \in \mathcal{D}} (y + \omega - (\theta + \widehat{\theta})^\top x)^2 + \frac{\sigma_\omega^2}{\sigma_p^2} \|\theta\|_2^2, \tag{B.4}$$

*could yield a sample from the posterior distribution $\mathcal{N}(\mathbb{E}[\theta^\star \mid \mathcal{D}], \mathrm{Cov}[\theta^\star \mid \mathcal{D}])$ defined by (B.1) and (B.2).*

First, it is obvious that (B.3) and (B.4) are equivalent by the variable change trick. Let us focus on (B.4) to explain the main idea of Lemma 2. Notice that the posterior sample includes two parts: the prior $\widehat{\theta}$ and the differential $\theta$ ($\theta$ is a specific optimal solution to (B.4)). In particular, $\theta$ is a random variable, which depends on the randomness in $\omega$. Based on this observation, we can leverage the

optimality condition to obtain the optimal solution of (B.4). Subsequently, we can verify that the mean of $\widehat{\theta} + \theta$ and its covariance are identical with the posterior mean $\mathbb{E}[\theta^\star \mid \mathcal{D}]$ and posterior covariance $\mathrm{Cov}[\theta^\star \mid \mathcal{D}]$, respectively.

**Remark 4.** *Importantly, Lemma 2 recasts a sampling problem to an optimization problem, which provides a purely computational cue to obtain a posterior sample. Consequently, it provides another way of implementing RLSVI by solving the least square problem in each iteration. However, there is no clear advantage for this implementation under the case where RLSVI can work (e.g., tabular or linear MDPs).*

## B.2 HYPERMODEL

In this part, we provide more explanation about the hypermodel. To proceed, note that the main issue of the procedure in Lemma 2 is its computational efficiency. The reason is that we have to solve multiple randomized least square problems if we want to obtain many posterior samples. One of the motivations of the hypermodel is to address this computational issue. Indeed, the original optimization problem in (Dwaracherla et al., 2020) is

$$\min_\nu \int_z p(z) \left[ \sum_{(x,y,\xi) \in \mathcal{D}} \left( y + \sigma_\omega z^\top \xi - g_{f_\nu(z)}(x) \right)^2 + \frac{\sigma_\omega^2}{\sigma_p^2} \left\| f_\nu(z) - f_{\nu_{\mathrm{prior}}}(z)(x) \right\|^2 \right] (\mathrm{d}z), \quad \text{(B.5)}$$

where $\nu_{\mathrm{prior}} = \nu_0$ is the fixed prior hypermodel parameter. Consider the linear case where $g_{f_\nu(z)}(x) = f_\nu(z)^\top x$, then (B.5) becomes

$$\min_\nu \int_z p(z) \left[ \sum_{(x,y,\xi) \in \mathcal{D}} \left( y + \sigma_\omega z^\top \xi - f_\nu(z)^\top x \right)^2 + \frac{\sigma_\omega^2}{\sigma_p^2} \left\| f_\nu(z) - f_{\nu_0}(z) \right\|^2 \right] (\mathrm{d}z),$$

which is very similar to (B.3) and the difference is discussed in Remark 1. Intuitively, the hypermodel wants to solve "infinite" randomized least-square problems simultaneously. Through this, the hypermodel can obtain multiple posterior samples by sampling $z$.

Note that (B.5) has a different form with the loss function in (4.1). However, as discussed in (Dwaracherla et al., 2020, Section 2.5), these two loss functions are equivalent. To derive (4.1) from (B.5), we take three steps: 1) replace $f_\nu(z)$ in (B.5) with $f_{\nu_{\mathrm{prior}}}(z) + f_{\nu'}(z)$; 2) let $g_{f_\nu(z)}(x) = g_{f_{\nu_{\mathrm{prior}}}(z) + f_{\nu'}(z)}(x) = g_{f_{\nu_{\mathrm{prior}}}(z)}(x) + g_{f_{\nu'}(z)}(x)$; 3) change the optimization variable from $\nu$ to $\nu'$.

## B.3 PROOF OF THEOREM 1

Here, we consider the linear hypermodel, i.e., $f_\nu(z) = \nu_w^\top z + \nu_b$, where $\nu_w \in \mathbb{R}^{N_z \times d}$ and $\nu_b \in \mathbb{R}^d$. By the property of Gaussian distribution, we know that $f_\nu(z) \sim \mathcal{N}(\nu_b, \nu_w^\top \nu_w)$ as $z \sim \mathcal{N}(0, I)$. Therefore, $\nu_b$ and $\nu_w^\top \nu_w$ qualify the mean and covariance, respectively.

By the formulation, there are two models in a complete hypermodel: the prior one and the differential one. Specifically,

- The prior hypermodel is
$$f_{\widehat{\nu}}(z) = \widehat{\nu}_w^\top z + \widehat{\nu}_b,$$
where $\widehat{\nu} = (\widehat{\nu}_w, \widehat{\nu}_b)$ is fixed and a non-trainable variable. Here, we set the $\widehat{\nu}_w^\top = \sigma_p I$ and $\widehat{\nu}_b = \bar{\theta}_p$ as the correct prior parameters, then $f_{\widehat{\nu}}(z)$ is a sample from the prior distribution $\mathcal{N}(\bar{\theta}_p, \sigma_p^2 I)$.

- The differential hypermodel is
$$f_\nu(z) = \nu_w^\top z + \nu_b,$$
where $\nu = (\nu_w, \nu_b)$ is a trainable variable.

Let the sample $(x_i, y_i)$ generated under Assumption 1. We augment each data sample $(x_i, y_i) \in \mathcal{D}$ with $\xi_i$ independently sampled from unit hypersphere (refer to Appendix A.3). Then, the dataset becomes

$$\mathcal{D} = \{(x_i, y_i, \xi_i) : i = 1, 2, \ldots, |\mathcal{D}|\},$$

with the augmented noise realizations

$$\Xi = \{\xi_i : i = 1, \ldots, |\mathcal{D}|\}.$$

**Theorem 1** (Formal statement). *Under Assumption 1, set $\nu_w^{prior} = \sigma_p I$ and $\nu_b^{prior} = \bar{\theta}_p$. Let $\nu^\star = (\nu_w^\star, \nu_b^\star)$ be the optimal solution of (4.1) conditioned on specific realizations of $\xi$, then $\theta := f_{\nu_{prior}}(z) + f_{\nu^\star}(z) \sim \mathcal{N}(\nu_b^{prior} + \nu_b^\star, (\nu_w^{prior} + \nu_w^\star)^\top(\nu_w^{prior} + \nu_w^\star))$ with*

$$\nu_b^{prior} + \nu_b^\star = \mathbb{E}[\theta^\star \mid \mathcal{D}], \quad (\nu_w^{prior} + \nu_w^\star)^\top(\nu_w^{prior} + \nu_w^\star) = \mathrm{Cov}[\theta^\star \mid \mathcal{D}] + err(\Xi),$$

*where*

$$err(\Xi) := \mathrm{Cov}[\theta^\star \mid \mathcal{D}]\left(\frac{1}{\sigma_\omega^2}\sum_{(x,\xi)\neq(x',\xi')\in\mathcal{D}} x\xi^\top\xi'x'^\top + \frac{1}{\sigma_\omega\sigma_p}\sum_{(x,\xi)\in\mathcal{D}}(x\xi^\top + \xi x^\top)\right)\mathrm{Cov}[\theta^\star \mid \mathcal{D}]$$

*Furthermore, the error term satisfies $\mathbb{E}_\Xi[err(\Xi)] = 0$.*

*Proof of Theorem 1.* For simplicity, we omit the subscript $^\star$ for the optimal solution to (4.1) in the proof. That is, we use the shorthand notation $\nu = (\nu_\omega, \nu_b)$ for the optimal solution to (4.1).

The objective function (4.1) for learning hypermodel given the data $\mathcal{D}$ becomes:

$$\mathcal{L}(\nu; \mathcal{D}) = \int_z p(z)\left(\sum_{(x,y,\xi)\in\mathcal{D}}\left(y + \sigma_\omega z^\top\xi - x^\top((\nu_w + \widehat{\nu}_w)^\top z + (\nu_b + \widehat{\nu}_b))\right)^2 + \frac{\sigma_\omega^2}{\sigma_p^2}\left\|\nu_w^\top z + \nu_b\right\|^2\right)(\mathrm{d}z)$$

According to the first-order optimality condition:

$$\frac{\partial\mathcal{L}}{\partial\nu_b} = \mathbb{E}_z\left[\sum_{(x,y,\xi)\in\mathcal{D}}(-x)\left(y + \sigma_\omega\xi^\top z - x^\top((\nu_w + \widehat{\nu}_w)^\top z + (\nu_b + \widehat{\nu}_b))\right) + \frac{\sigma_\omega^2}{\sigma_p^2}(\nu_w^\top z + \nu_b) \mid \mathcal{D}\right] \tag{B.6}$$

$$= \sum_{(x,y,\xi)\in\mathcal{D}} x(x^\top\nu_b + x^\top\widehat{\nu}_b - y) + \frac{\sigma_\omega^2}{\sigma_p^2}\nu_b = 0.$$

It is straightforward to obtain that

$$\sum_{(x,y,\xi)\in\mathcal{D}} x(x^\top\nu_b + x^\top\widehat{\nu}_b - y) + \frac{\sigma_\omega^2}{\sigma_p^2}(\nu_b + \widehat{\nu}_b) = \frac{\sigma_\omega^2}{\sigma_p^2}\widehat{\nu}_b.$$

Then, we can infer that

$$\nu_b + \widehat{\nu}_b = \left(\frac{1}{\sigma_\omega^2}\sum_{x\in\mathcal{D}} xx^\top + \frac{1}{\sigma_p^2}I\right)^{-1}\left(\frac{1}{\sigma_\omega^2}\sum_{(x,y)\in\mathcal{D}} xy + \frac{1}{\sigma_p^2}\widehat{\nu}_b\right)$$

$$= \left(\frac{1}{\sigma_\omega^2}X^\top X + \frac{1}{\sigma_p^2}I\right)^{-1}\left(\frac{1}{\sigma_\omega^2}X^\top Y + \frac{1}{\sigma_p^2}\bar{\theta}_p\right)$$

$$= \mathbb{E}\left[\theta^\star \mid \mathcal{D}\right],$$

where $X \in \mathbb{R}^{|\mathcal{D}|\times d}$, $Y \in \mathbb{R}^{|\mathcal{D}|}$, and $\mathbb{E}\left[\theta \mid \mathcal{D}\right]$ is defined in (B.1). This implies that the hypermodel can recover the posterior mean.

For the variable $\nu_w$, we calculate its partial derivative $\partial\mathcal{L}/\partial\nu_w^\top$ as

$$\mathbb{E}_z\left[\sum_{(x,y,\xi)\in\mathcal{D}}\left(y + \sigma_\omega z^\top\xi - x^\top((\nu_w + \widehat{\nu}_w)^\top z + (\nu_b + \widehat{\nu}_b))\right)(-xz^\top) + \frac{\sigma_\omega^2}{\sigma_p^2}(\nu_w^\top z + \nu_b)z^\top \mid \mathcal{D}\right]$$

$$= \mathbb{E}_z\left[\sum_{(x,y,\xi)\in\mathcal{D}}\left(-\sigma_\omega z^\top\xi xz^\top + x^\top(\nu_w + \widehat{\nu}_w)^\top zxz^\top\right) + \frac{\sigma_\omega^2}{\sigma_p^2}\nu_w^\top zz^\top \mid \mathcal{D}\right]. \tag{B.7}$$

To proceed, we utilize the following helpful lemma.

**Lemma 3.** *Let $z \sim \mathcal{N}(0, I_d)$. For any fixed vector $a \in \mathbb{R}^d$, we have*

$$\mathbb{E}_z\left[a^\top zxz^\top\right] = xa^\top.$$

*Proof.* The proof is easy if we look at each entry of the matrix,

$$\left[\mathbb{E}_z\left[z^\top a x z^\top\right]\right]_{ij} = \mathbb{E}_z\left[z^\top a[xz^\top]_{ij}\right] = \mathbb{E}_z\left[\left(\sum_{k=1}^d a_k z_k\right) x_i z_j\right] = a_j x_i = [xa^\top]_{ij}.$$

□

Then, with the above Lemma 3 and (B.7), we further calculate

$$\frac{\partial\mathcal{L}}{\partial\nu_w^\top} = \sum_{(x,y,\xi)\in\mathcal{D}}\left(-\sigma_\omega x\xi^\top + xx^\top(\nu_w + \widehat{\nu}_w)\right) + \frac{\sigma_\omega^2}{\sigma_p^2}\nu_w^\top. \tag{B.8}$$

By the first order optimality condition, we have

$$(\nu_w + \widehat{\nu}_w) = \left(\frac{1}{\sigma_\omega^2}X^\top X + \frac{1}{\sigma_p^2}I\right)^{-1}\left(\frac{1}{\sigma_\omega^2}\sum_{(x,\xi)\in\mathcal{D}}\sigma_\omega x\xi^\top + \frac{1}{\sigma_p^2}\widehat{\nu}_w\right).$$

With the posterior covariance in (B.2)

$$\Sigma^{-1} := \text{Cov}(\theta^\star \mid \mathcal{D}) = \left(\frac{1}{\sigma_\omega^2}X^\top X + \frac{1}{\sigma_p^2}I\right)^{-1},$$

and define the set $\Xi = (\xi_1, \ldots, \xi_{|\mathcal{D}|})$ corresponding to the $\xi_i$ in $(x_i, y_i, \xi_i) \in \mathcal{D}$, we obtain that

$$(\nu_w + \widehat{\nu}_w)^\top(\nu_w + \widehat{\nu}_w)$$

$$= \Sigma^{-1}\left(\frac{1}{\sigma_\omega^2}\sum_{(x,\xi)}\sum_{(x',\xi')}x\xi^\top\xi'x'^\top + \frac{1}{\sigma_p^4}\widehat{\nu}_w^\top\widehat{\nu}_w + \frac{1}{\sigma_\omega^2\sigma_p^2}\sum_{(x,\xi)}\left(\sigma_\omega x\xi^\top\widehat{\nu}_w + \sigma_\omega\widehat{\nu}_w^\top\xi x^\top\right)\right)\Sigma^{-1}$$

$$= \Sigma^{-1}\left(\frac{1}{\sigma_\omega^2}X^\top X + \frac{1}{\sigma_p^2}I + \frac{1}{\sigma_\omega^2}\sum_{(x,\xi)\neq(x',\xi')}x\xi^\top\xi'x'^\top + \frac{1}{\sigma_\omega\sigma_p}\sum_{(x,\xi)}(x\xi^\top + \xi x^\top)\right)\Sigma^{-1}$$

$$= \Sigma^{-1}\Sigma\Sigma^{-1} + \text{err}(\Xi),$$

where the second equality is due to the fact that $\|\xi\| = 1$ and we set $\widehat{\nu}_w = \sigma_p I$ at the beginning; and $\text{err}(\Xi) = \Sigma^{-1}\left(\frac{1}{\sigma_\omega^2}\sum_{(x,\xi)\neq(x',\xi')}x\xi^\top\xi'x'^\top + \frac{1}{\sigma_\omega\sigma_p}\sum_{(x,\xi)}(x\xi^\top + \xi x^\top)\right)\Sigma^{-1}$. Taking the expectation over $\Xi = (\xi_1, \ldots, \xi_{|\mathcal{D}|})$, we have

$$\mathbb{E}_\Xi\left[\sum_{(x,\xi)\neq(x',\xi')}x\xi^\top\xi'x'^\top\right] = 0, \mathbb{E}_\Xi\left[\sum_{(x,\xi)}(x\xi^\top + \xi^\top x)\right] = 0.$$

Finally, we have

$$\mathbb{E}_\Xi\left[(\nu_w + \widehat{\nu}_w)^\top(\nu_w + \widehat{\nu}_w)\right] = \Sigma^{-1}\Sigma\Sigma^{-1} + \mathbb{E}_\Xi[\text{err}(\Xi)] = \Sigma^{-1}.$$

This implies the hypermodel can also recover the posterior covariance in expectation. □

## B.4 WHY INDEPENDENT GAUSSIAN NOISE CANNOT WORK FOR HYPERMODEL?

Following similar steps in Appendix B.3, we want to argue that the posterior approximation result cannot be achieved if we replace $z^\top\xi$ with an independent Gaussian noise $\omega$ in (4.1).

Similarly, we augment each sample $(x_i, y_i) \in \mathcal{D}$ with $\omega_i \sim \mathcal{N}(0, \sigma_\omega^2)$ and denote $\boldsymbol{\omega} = (\omega_1, \omega_2, \ldots, \omega_{|\mathcal{D}|})^\top$ as the noise vector. The dataset becomes

$$\mathcal{D} = (x_i, y_i, \omega_i : i = 1, 2, \ldots, |\mathcal{D}|).$$

Now, the objective function for learning hypermodel given the data $\mathcal{D}$ becomes:

$$\mathcal{L}^\omega(\nu; \mathcal{D}) = \int_z p(z)\left(\sum_{(x,y,\omega)\in\mathcal{D}}\left(y + \omega - x^\top((\nu_w + \widehat{\nu}_w)^\top z + (\nu_b + \widehat{\nu}_b))\right)^2 + \frac{\sigma_\omega^2}{\sigma_p^2}\left\|\nu_w^\top z + \nu_b\right\|^2\right)(\text{d}z).$$

Similar to (B.6), we have the first order optimality condition

$$\frac{\partial \mathcal{L}^\omega}{\partial \nu_b} = \sum_{(x,y,\omega)} x(x^\top b + x^\top \widehat{\nu}_b - y + \omega) + \frac{\sigma_\omega^2}{\sigma_p^2} \nu_b = 0.$$

Then, we have

$$\nu_b + \widehat{\nu}_b = \left( \frac{1}{\sigma_\omega^2} X^\top X + \frac{1}{\sigma_p^2} I \right)^{-1} \left( \frac{1}{\sigma_\omega^2} X^\top (Y + \omega) + \frac{1}{\sigma_p^2} \bar{\theta}_p \right) \neq \mathbb{E}\left[ \theta \mid \mathcal{D} \right].$$

For the variable $\nu_w$, similar to (B.7), we calculate the partial derivative $\partial \mathcal{L} / \partial \nu_w^\top$ as

$$\mathbb{E}_z \left[ \sum_{(x,y,\xi) \in \mathcal{D}} \left( y + \omega - x^\top ((\nu_w + \widehat{\nu}_w)^\top z + (\nu_b + \widehat{\nu}_b)) \right) (-xz^\top) + \frac{\sigma_\omega^2}{\sigma_p^2} (\nu_w^\top z + \nu_b) z^\top \mid \mathcal{D} \right]$$

$$= \mathbb{E}_z \left[ \sum_{(x,y,\omega) \in \mathcal{D}} \left( -\omega x z^\top + x^\top (\nu_w + \widehat{\nu}_w)^\top z x z^\top \right) + \frac{\sigma_\omega^2}{\sigma_p^2} \nu_w^\top z z^\top \mid \mathcal{D} \right]$$

$$= \mathbb{E}_z \left[ \sum_{(x,y,\omega) \in \mathcal{D}} \left( x^\top (\nu_w + \widehat{\nu}_w)^\top z x z^\top \right) + \frac{\sigma_\omega^2}{\sigma_p^2} \nu_w^\top z z^\top \mid \mathcal{D} \right]$$

$$= \sum_{(x,y,\omega) \in \mathcal{D}} \left( x x^\top (\nu_w + \widehat{\nu}_w)^\top \right) + \frac{\sigma_\omega^2}{\sigma_p^2} \nu_w^\top.$$

By the first order optimality condition, we have

$$(\nu_w + \widehat{\nu}_w)^\top = \left( \frac{1}{\sigma_\omega^2} X^\top X + \frac{1}{\sigma_p^2} I \right)^{-1} \left( \frac{1}{\sigma_p^2} \widehat{\nu}_w^\top \right).$$

Therefore, by the fact that $\widehat{\nu}_w^\top \widehat{\nu}_w = \sigma_p^2 I$,

$$(\nu_w + \widehat{\nu}_w)^\top (\nu_w + \widehat{\nu}_w) = \Sigma^{-1} \left( \frac{1}{\sigma_p^2} I \right) \Sigma^{-1} = \frac{1}{\sigma_p^2} \Sigma^{-2} \neq \mathrm{Cov}\left( \theta \mid \mathcal{D} \right).$$

This implies that an independent Gaussian noise $\omega$ cannot work and the $z$-dependent noise is indispensable for posterior approximation.

## C  EXPERIMENT DETAILS

### C.1  ALGORITHM IMPLEMENTATION AND PARAMETERS

**Common Hyperparameters.** All agents use the same network structure as in (Mnih et al., 2015) on Atari and SuperMarioBros:

$$\text{state} \rightarrow \mathbf{conv}(32, 8, 4) \rightarrow \mathbf{relu} \rightarrow \mathbf{conv}(64, 4, 2) \rightarrow \mathbf{relu} \rightarrow \mathbf{conv}(64, 3, 1)$$
$$\rightarrow \mathbf{mlp}(512) \rightarrow \mathbf{relu} \rightarrow \mathbf{mlp}(\text{number of actions}),$$

where $\mathbf{conv}(32, 8, 4)$ means a convolution layer with 64 filters of size 8 and stride of 4, and $\mathbf{mlp}(512)$ means a fully-connected layer with output size of 512, and $\mathbf{relu}$ stands for Rectified Linear Units.

The algorithmic parameters on Atari and SuperMarioBros basically follow (Mnih et al., 2015, Table 1). For example, the replay buffer size is 1M; the batch size is 32; the discount factor is 0.99; the target network update frequency is 10K agent steps; the train frequency is 4 agent steps; and the replay starts after 50K agent steps. For algorithms with $\epsilon$-greedy (e.g., DQN, BootDQN, OPIQ and OB2I), the exploration $\epsilon$ is annealed from 1.0 to 0.1 linearly (from 50K agent steps to 1M agent steps, respectively); the test $\epsilon$ is 0.05.

**DQN.** The training result of DQN (Mnih et al., 2015) on Atari is based on DQN Zoo (Quan & Ostrovski, 2020)[9]. We implement DQN with `tianshou`[10] framework and train it on SuperMarioBros.

---

[9]https://github.com/deepmind/dqn_zoo/blob/master/results.tar.gz
[10]https://github.com/thu-ml/tianshou

**OPIQ.** We use the implementation in the public repository `https://github.com/oxwhirl/opiq`. For a fair comparison, we do no use the mixed Monte Carlo return and set the final $\epsilon$ to be 0.05. Other parameters follow (Rashid et al., 2020).

**OB2I.** We use the implementation for OB2I in the public repository `https://github.com/Baichenjia/OB2I`. The training data of OB2I on Atari is partially shared by authors in (Bai et al., 2021) (private communication). All parameters follow (Bai et al., 2021).

**BootDQN.** We modify the implementation for BootDQN from the public repository `https://github.com/johannah/bootstrap_dqn` to make a fair comparison. In particular, BootDQN uses 10 independent ensembles and the $\epsilon$-greedy strategy is same with DQN. We use the version with prior value functions (Osband et al., 2018). Note that we do not implement the "vote" mode because we observe that it does not matter in practice.

**NoisyNet.** We modify the implementation for NoisyNet from the public repository `https://github.com/ku2482/fqf-iqn-qrdqn.pytorch/blob/master/fqf_iqn_qrdqn/network.py` to make a fair comparison. In particular, NoisyNet re-samples a noisy network for action selection and update. The noise scale is 0.5 and other parameters follow (Fortunato et al., 2018).

**HyperDQN.** We use the implementation as described in Appendix A.2. Algorithm parameters are listed in Table 4. To stabilize training, each $z$ corresponds to 32 mini-batch samples and the effective batch size of HyperDQN is $32 \times 10 = 320$.

Table 4: Algorithmic parameters of HyperDQN for Atari and SuperMarioBros.

|  | Atari | SuperMarioBros | Deep Sea |
|---|---|---|---|
| $z$ dimension $N_z$ | 32 | 32 | 2 |
| prior scale $\beta_{\text{prior}}$ in (A.5) | 0.1 | 0.1 | 10.0 |
| differential scale $\beta_{\text{differential}}$ in (A.5) | 0.1 | 0.1 | 1.0 |
| number of $z$ for training in (A.6) | 10 | 10 | 20 |
| noise scale $\sigma_w$ in (A.6) | 0.01 | 0.01 | 0.0 |
| prior regularization scale $\sigma_p$ in (A.6) | 0.1 | 0.1 | 0.0 |
| learning rate | 0.0001 | 0.0001 | 0.001 |

## C.2 ENVIRONMENT PREPROCESSING

**Atari.** We follow the same preprocessing as in (Mnih et al., 2015) and (van Hasselt et al., 2016). In particular, there are random no-operation steps (up to 30) before the interaction. Each agent step corresponds to 4 environment steps by repeating the same action while each environment step corresponds to 4 frames of the simulator. The raw score is clipped to $\{-1, 0, +1\}$ for training but the evaluation performance is based on the raw score. Episodes are early stopped after 108K frames as in (van Hasselt et al., 2016). The observation for the agent is based on 4 stacked frames, which is reshaped to $(4, 84, 84)$.

**SuperMarioBros.** The observation and action preprocessing are the same with Atari 2600 suite. Two things are different: 1) there is no random "no operation"; 2) the training reward is based on $0.01 \times$ `raw score` rather than clipping. Moreover, the maximum episode length is $1500 \times 4 = 6000$ frames.

For algorithms (DQN, BootDQN, NoisyNet, HyperDQN) that we implement by the `tianshou` framework, the training frequency is 10 and the target update frequency is 500 to accelerate training speed. Other parameters are identical to the one for Atari.

**Deep sea.** The implementation of the deep sea task is from `bsuite`[11] (Osband et al., 2020). The neural network architecture is :

$$\text{state} \rightarrow \mathbf{mlp}(64) \rightarrow \mathbf{relu} \rightarrow \mathbf{mlp}(64) \rightarrow \mathbf{relu} \rightarrow \mathbf{mlp}(\text{number of actions}).$$

---

[11]`https://github.com/deepmind/bsuite/blob/master/bsuite/environments/deep_sea.py`

To make a fair comparison with BootDQN, we use the setting in (Osband et al., 2018). In particular, the training frequency is 1, target update frequency is 4, batch size is 128, and buffer size is 200K. For HyperDQN, its parameters are listed in Table 4. For BootDQN, it uses 10 ensembles and the other parameters are identical to HyperDQN. Both algorithms do not use $\epsilon$-greedy.

### C.3 BAYESIAN LINEAR REGRESSION

The 2-dimensional Bayesian linear regression problem used in Section 4.3 is based on $y = \theta_w \cdot x + \theta_b + \epsilon$, where $x, y, \theta_w, \theta_b, \epsilon \in \mathbb{R}$. Specifically, $\epsilon$ is sampled from the Gaussian distribution. To generate the dataset, $x$ is sampled from $[-4, 4]$ uniformly. The total number of training samples is 50. The prior distribution for $(\theta_w, \theta_b)$ is $\mathcal{N}(0, I_2)$ while the actual value $(\theta_w, \theta_b)$ is $(6.06, 0.47)$ for the dataset generation.

We obtain the exact posterior distribution by the Bayes update rule. For hypermodel, the dimension of $z$ is 2. We use gradient descent with a learning rate of $0.005$ and momentum of $0.9$. The number of of the gradient descent iterations is $10000$. After the optimization, we visualize the 5000 samples from different distributions (e.g., the posterior distribution and the one transformed by the hypermodel) in Figure 2. Indeed, the KL-divergence between the exact posterior distribution and the counterparts by `hypermodel($z^\top\xi$)` and `hypermodel($\omega$)` are $0.1$ and $326.4$, respectively.

### C.4 PARAMETER CHOICE

**Noise scale $\sigma_\omega$.** In this part, we provide the ablation study about the noise scale $\sigma_\omega$. The numerical result is displayed in Figure 7. We observe the performance of HyperDQN is not very sensitive to $\sigma_\omega$.

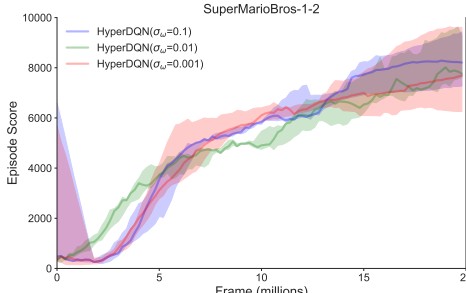

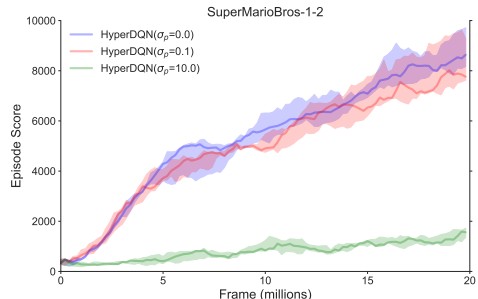

Figure 7: Comparison of HyperDQN with different noise scales $\sigma_\omega$.

Figure 8: Comparison of HyperDQN with different prior scales $\sigma_p$.

**Prior scale $\sigma_p$.** In this part, we provide the ablation study about the prior scale $\sigma_p$. The numerical result is shown in Figure 8. We find that a large prior scale results in a poor performance. The reason is that the posterior update is slow under this case.

**Hypermodel architecture.** In this paper, we mainly focus on the linear hypermodel. Here, we investigate the variant with a non-linear hypermodel (i.e., an MLP hypermodel). In particular, we consider the hypermodel is a two-layer neural network of width 64 with ReLU activation. We call this variant as `HyperDQN(MLP)` and the original HyperDQN as `HyperDQN(Linear)`. It is intuitive that `HyperDQN(MLP)` has a more powerful posterior approximation ability but it may be hard to train `HyperDQN(MLP)` since the architecture is more complex. The empirical results on Atari and SuperMarioBros are shown in Figure 9. We see that HyperDQN does not obtain significant gains by an MLP hypermodel. We conjecture the reason is the trainability issue of the MLP hypermodel.

**Number of ensembles in BootDQN.** In this paper, we implement BootDQN with 10 ensembles, which follows the configuration in (Osband et al., 2016a, Section 6.1). However, in (Osband et al., 2018), BootDQN is implemented with 20 ensembles. We remark that the choice of 10 ensembles is commonly used in the previous literature (Rashid et al., 2020; Bai et al., 2021) since it is more computationally cheap. For completeness, we provide the ablation study about this parameter choice; see Figure 10 for the result.

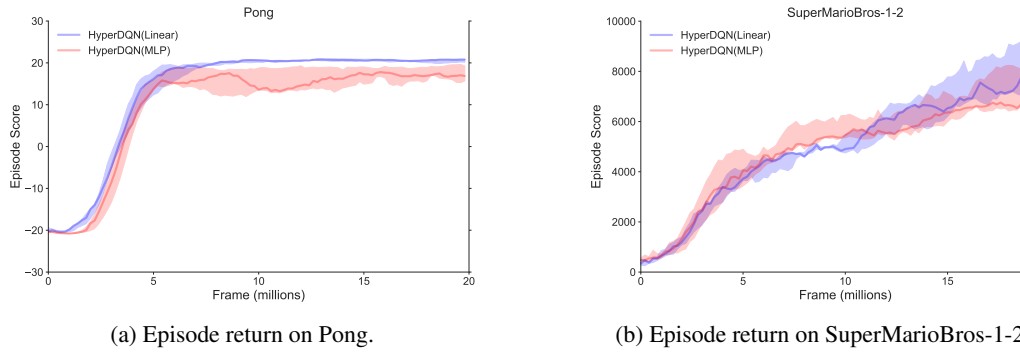

(a) Episode return on Pong.  (b) Episode return on SuperMarioBros-1-2.

Figure 9: Comparison of HyperDQN with a linear hypermodel and a MLP hypermodel.

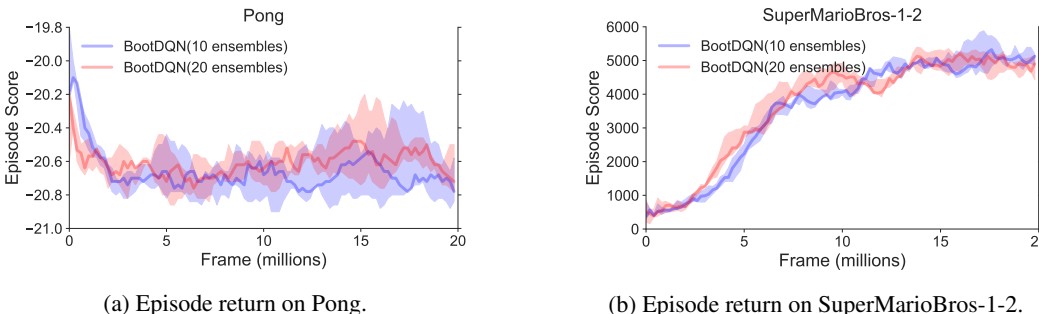

(a) Episode return on Pong.  (b) Episode return on SuperMarioBros-1-2.

Figure 10: Comparison of BootDQN with 10 ensembles and 20 ensembles.

## D  ADDITIONAL RESULTS

### D.1  ATARI

For all algorithms, learning curves on each game are visualized in Figure 11. The maximum raw scores on each individual game are reported in Table 5.

**Relative improvement on each game.** To better understand the improvement of HyperDQN, we visualize the relative score compared with baselines. Specifically, the relative score is calculated as $\frac{proposed-baseline}{\max(human,baseline)-human}$ (Wang et al., 2016). According to the taxonomy in (Bellemare et al., 2016), we cluster environments by 4 groups: "hard exploration (dense reward)", "hard exploration (sparse reward)", "easy exploration", and "unknown". See the results in Figure 12, Figure 13, Figure 14, and Figure 15.

We observe that HyperDQN has improvements on both "easy exploration" environments (e.g., Battle Zone, Jamesbond, and Pong) and "hard exploration" environments (e.g., Frostbite, Gravitar, and Zaxxon). However, we notice that HyperDQN does not work well on very sparse reward tasks like Montezuma's Revenge. We explain the failure reason as follows. As we have discussed in the introduction, without a good feature, randomized exploration methods are rarely competent. On Montezuma's Revenge, the extremely sparse reward provides limited feedback for feature selection. As a result, it is expected that randomized exploration methods (including BootDQN and NoisyNet) do not work well for this task. In contrast, prediction error based methods (Pathak et al., 2017; Burda et al., 2019b; Rashid et al., 2020) could leverage specific architecture designs to provide auxiliary reward feedback to help feature selection and exploration. As a result, these methods perform well on Montezuma's Revenge. We kindly remind that these methods do not perform well on other tasks because specifically designed architectures in these methods do not generalize well as pointed out in (Taïga et al., 2020).

**Commitment of randomized exploration.** Here we provide the evidence that using the $\epsilon$-greedy strategy contradicts the commitment and leads to poor performance for HyperDQN; see the results in Figure 16. We observe that combing HyperDQN with $\epsilon$-greedy results in poor performance.

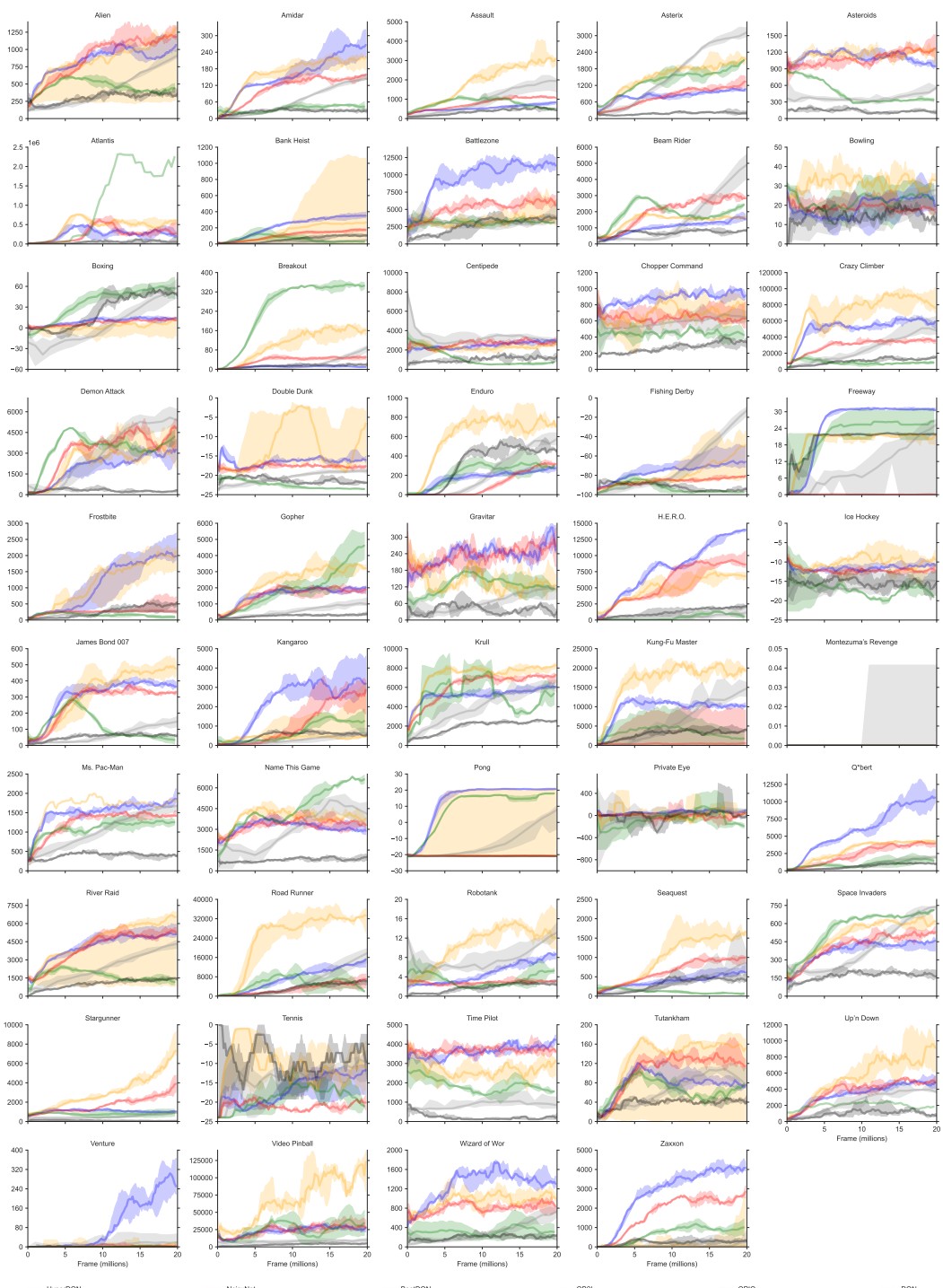

Figure 11: Learning curves of algorithms on Atari. Solid lines correspond to the median performance over 3 random seeds while shaded ares correspond to 90% confidence interval. Same with other figures for Atari.

## D.2 SUPERMARIOBROS

For all algorithms, learning curves on each game are visualized in Figure 17.

Table 5: The maximal score over 200 evaluation episodes for the best policy in hindsight (after 20M frames) for Atari games. The performance of the random policy and the human expert is from https://github.com/deepmind/dqn_zoo/blob/master/dqn_zoo/atari_data.py#L41-L101. The performance of OB2I is from (Bai et al., 2021).

| | Random | Human | OB2I | OPIQ | BootDQN | NoisyNet | HyperDQN |
|---|---|---|---|---|---|---|---|
| Alien | 227.8 | 7,127.7 | 916.9 | 2,316.7 | 2,623.3 | 2,596.7 | 2,910.0 |
| Amidar | 5.8 | 1,719.5 | 94.0 | 161.6 | 319.0 | 395.7 | 565.7 |
| Assault | 222.4 | 742.0 | 2,996.2 | 3,385.0 | 3,182.7 | 5,000.3 | 2,083.0 |
| Asterix | 210.0 | 8,503.3 | 2,719.0 | 1,500.0 | 3,466.7 | 4,000.0 | 2,283.3 |
| Asteroids | 719.1 | 47,388.7 | 959.9 | 976.7 | 3,353.3 | 3,143.3 | 3,190.0 |
| Atlantis | 12,850.0 | 29,028.1 | 3,146,300.0 | 1,780,266.7 | 835,166.7 | 857,100.0 | 880,233.3 |
| Bank Heist | 14.2 | 753.1 | 378.6 | 430.0 | 270.0 | 250.0 | 470.0 |
| Battle Zone | 2,360.0 | 13,454.5 | 8,756.5 | 18,333.3 | 22,666.7 | 20,333.3 | 24,333.3 |
| BeamRider | 363.9 | 16,926.5 | 3,736.7 | 4,385.3 | 7,002.0 | 5,467.3 | 3,955.3 |
| Bowling | 23.1 | 160.7 | 30.0 | 56.3 | 90.3 | 116.0 | 64.0 |
| Boxing | 0.1 | 12.1 | 75.1 | 96.0 | 60.7 | 53.7 | 56.3 |
| Breakout | 1.7 | 30.5 | 423.1 | 247.0 | 212.0 | 352.0 | 43.7 |
| Centipede | 2,090.9 | 12,017.0 | 2,661.8 | 11,891.7 | 11,051.7 | 9,492.3 | 9,923.0 |
| Chopper Command | 811.0 | 7,387.8 | 1,100.3 | 1,666.7 | 1,900.0 | 2,966.7 | 2,733.3 |
| Crazy Climber | 10,780.5 | 35,829.4 | 53,346.7 | 71,566.7 | 88,400.0 | 138,133.3 | 117,966.7 |
| Demon Attack | 152.1 | 1,971.0 | 6,794.6 | 3,805.0 | 9,148.3 | 8,845.0 | 8,755.0 |
| Double Dunk | -18.6 | -16.4 | -18.2 | -18.7 | -6.0 | 3.3 | 2.7 |
| Enduro | 0.0 | 860.5 | 719.0 | 1,033.3 | 626.3 | 1,169.7 | 456.0 |
| Fishing Derby | -91.7 | 5.5 | -60.1 | -91.3 | -54.3 | -3.7 | -26.3 |
| Freeway | 0.0 | 29.6 | 32.1 | 26.3 | 9.3 | 25.0 | 32.7 |
| Frostbite | 65.2 | 4,334.7 | 1,277.3 | 1,640.0 | 1,853.3 | 4,020.0 | 3,943.3 |
| Gopher | 257.6 | 2,412.5 | 6,359.5 | 2,266.7 | 5,466.7 | 7,606.7 | 4,600.0 |
| Gravitar | 173.0 | 3,351.4 | 393.6 | 450.0 | 966.7 | 1,250.0 | 1,316.7 |
| H.E.R.O | 1,027.0 | 30,826.4 | 3,302.5 | 8,345.0 | 13,590.0 | 12,675.0 | 20,156.7 |
| Ice Hockey | -11.2 | 0.9 | -4.2 | -11.7 | -4.0 | 0.7 | -1.7 |
| Jamesbond | 29.0 | 302.8 | 434.3 | 350.0 | 650.0 | 650.0 | 700.0 |
| Kangaroo | 52.0 | 3,035.0 | 2,387.0 | 2,400.0 | 8,666.7 | 2,333.3 | 10,166.7 |
| Krull | 1,598.0 | 2,665.5 | 45,388.8 | 3,763.3 | 10,643.3 | 10,260.0 | 8,413.3 |
| Kung-Fu Master | 258.5 | 22,736.3 | 16,272.2 | 18,033.3 | 1,500.0 | 34,933.3 | 26,933.3 |
| Montezuma's Revenge | 0.0 | 4,753.3 | 0.0 | 0.0 | 0.0 | 0.0 | 0.0 |
| Ms. Pacman | 307.3 | 6,951.6 | 1,794.9 | 2,186.7 | 3,793.3 | 4,440.0 | 4,590.0 |
| Name This Game | 2,292.3 | 4,076.0 | 8,576.8 | 4,883.3 | 8,200.0 | 9,093.3 | 7,106.7 |
| Pong | -20.7 | 14.6 | 18.7 | -20.0 | -17.7 | 2.5 | 21.0 |
| Private Eye | 24.9 | 69,571.3 | 1,174.1 | 5,124.3 | 7,641.0 | 11,018.7 | 2,810.0 |
| Q*Bert | 163.9 | 13,455.0 | 4,275.0 | 4,558.3 | 5,250.0 | 5,966.7 | 16,616.7 |
| River Raid | 1,338.5 | 17,118.0 | 2,926.5 | 4,536.7 | 8,003.3 | 5,993.3 | 8,020.0 |
| Road Runner | 11.5 | 7,845.0 | 21,831.4 | 20,866.7 | 17,266.7 | 3,533.3 | 27,033.3 |
| Robotank | 2.2 | 11.9 | 13.5 | 15.3 | 11.7 | 37.7 | 19.7 |
| Seaquest | 68.4 | 42,054.7 | 332.1 | 1,846.7 | 2,180.0 | 230.0 | 1,360.0 |
| Space Invaders | 148.0 | 1,668.7 | 904.9 | 853.3 | 1,390.0 | 1,410.0 | 925.0 |
| Star Gunner | 664.0 | 10,250.0 | 1,290.2 | 933.3 | 13,300.0 | 22,066.7 | 3,633.3 |
| Tennis | -23.8 | -8.3 | -1.0 | -20.3 | -1.0 | -1.0 | 5.3 |
| Time Pilot | 3,568.0 | 5,229.2 | 3,404.5 | 4,100.0 | 8,400.0 | 8,433.3 | 10,166.7 |
| Tutankham | 11.4 | 167.6 | 297.0 | 184.0 | 269.3 | 181.3 | 213.3 |
| Up and Down | 533.4 | 11,693.2 | 5,100.8 | 37,893.3 | 16,333.3 | 52,476.7 | 16,520.0 |
| Venture | 0.0 | 1,187.5 | 16.1 | 133.3 | 266.7 | 700.0 | 966.7 |
| Video Pinball | 16,256.9 | 17,667.9 | 80,607.0 | 53,924.7 | 262,718.0 | 881,999.3 | 172,896.3 |
| Wizard of Wor | 563.5 | 4,756.5 | 480.7 | 2,500.0 | 4,266.7 | 6,066.7 | 7,266.7 |
| Zaxxon | 32.5 | 9,173.3 | 2,842.0 | 3,300.0 | 8,400.0 | 4,233.3 | 10,066.7 |

**Commitment of randomized exploration.** Here we provide the evidence that using the $\epsilon$-greedy strategy contradicts the commitment and leads to poor performance for HyperDQN on SuperMario-Bros; see the result in Figure 18.

In addition, we provide the evidence that the randomized exploration approach BootDQN (Osband et al., 2016a) could obtain some gains if $\epsilon$-greedy is not used; see the result in Figure 19. The result suggests that the original implementation of BootDQN does not satisfy the commitment property. Note that the variant without $\epsilon$-greedy is still inferior compared with HyperDQN.

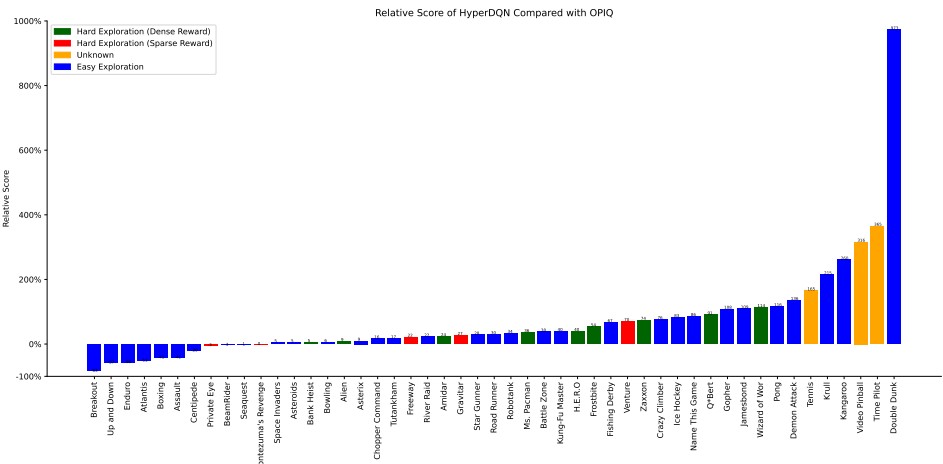

Figure 12: Relative improvement of HyperDQN compared with OPIQ (Rashid et al., 2020) on Atari. The relative performance is calculated as $\frac{\text{proposed}-\text{baseline}}{\max(\text{human},\text{baseline})-\text{human}}$ (Wang et al., 2016). Environments are grouped according to the taxonomy in (Bellemare et al., 2016, Table 1). "Unknown" indicates such environments are not considered in (Bellemare et al., 2016).

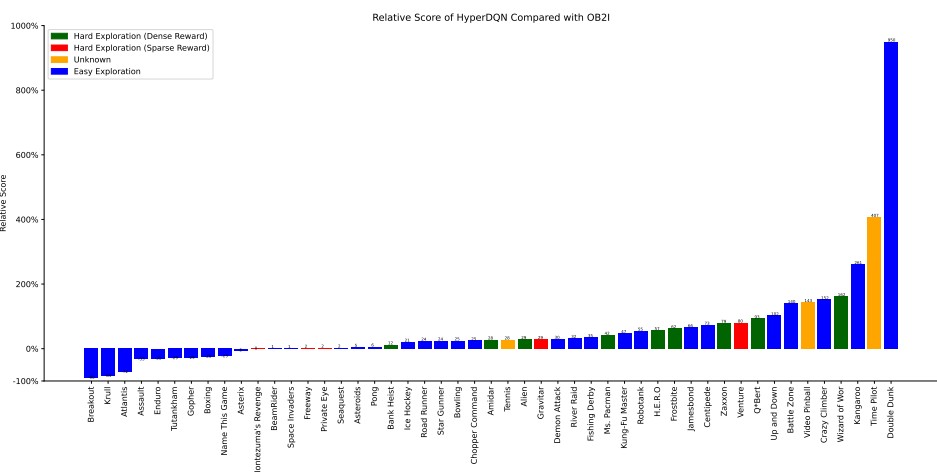

Figure 13: Relative improvement of HyperDQN compared with OB2I (Bai et al., 2021) on Atari. The relative performance is calculated as $\frac{\text{proposed}-\text{baseline}}{\max(\text{human},\text{baseline})-\text{human}}$ (Wang et al., 2016). Environments are grouped according to the taxonomy in (Bellemare et al., 2016, Table 1). "Unknown" indicates such environments are not considered in (Bellemare et al., 2016).

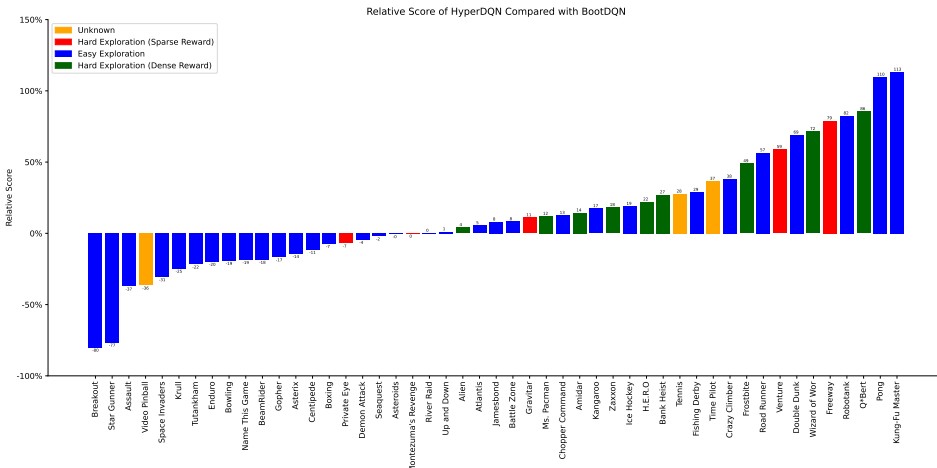

Figure 14: Relative improvement of HyperDQN compared with BootDQN (Osband et al., 2018) on Atari. The relative performance is calculated as $\frac{\text{proposed}-\text{baseline}}{\max(\text{human},\text{baseline})-\text{human}}$ (Wang et al., 2016). Environments are grouped according to the taxonomy in (Bellemare et al., 2016, Table 1). "Unknown" indicates such environments are not considered in (Bellemare et al., 2016).

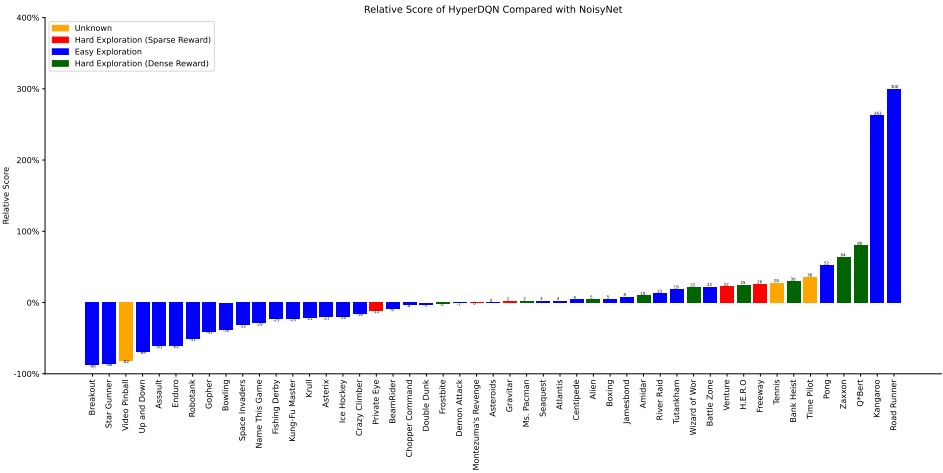

Figure 15: Relative improvement of HyperDQN compared with NoisyNet (Fortunato et al., 2018) on Atari. The relative performance is calculated as $\frac{\text{proposed}-\text{baseline}}{\max(\text{human},\text{baseline})-\text{human}}$ (Wang et al., 2016). Environments are grouped according to the taxonomy in (Bellemare et al., 2016, Table 1). "Unknown" indicates such environments are not considered in (Bellemare et al., 2016).

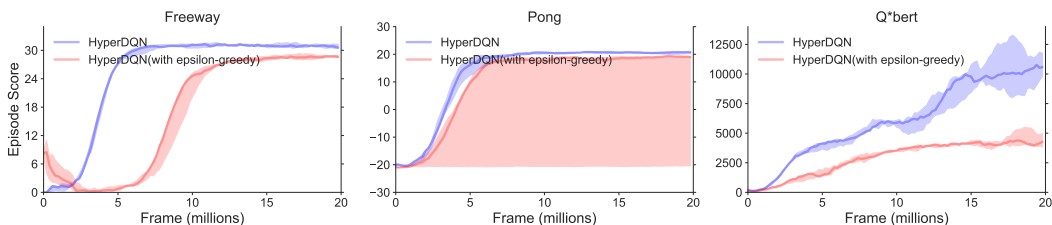

Figure 16: Comparison of HyperDQN with and without $\epsilon$-greedy on Atari.

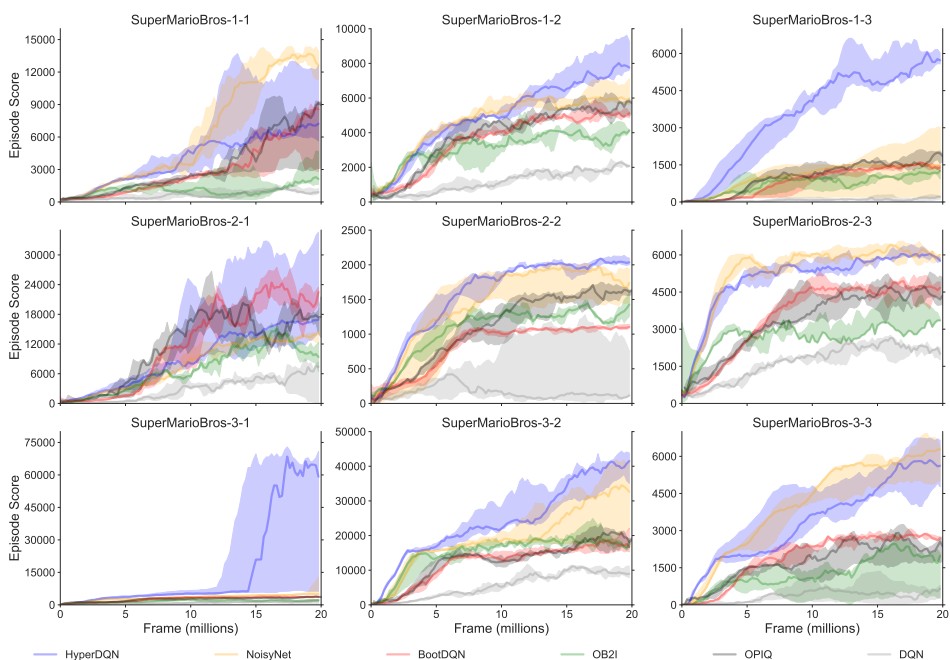

Figure 17: Learning curves of algorithms on SuperMarioBros. Solid lines correspond to the median performance over 3 random seeds while shaded ares correspond to 90% confidence interval. Same with other figures for SuperMarioBros.

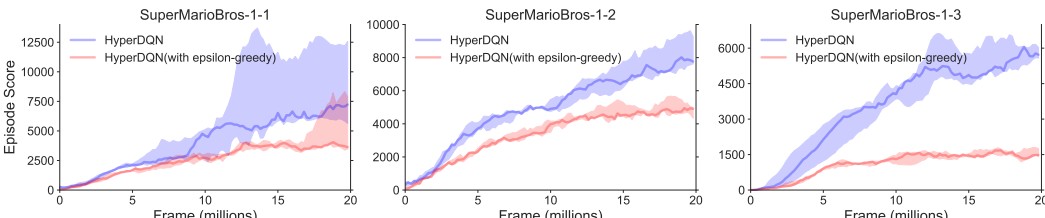

Figure 18: Comparison of HyperDQN with and without $\epsilon$-greedy on SuperMarioBros.

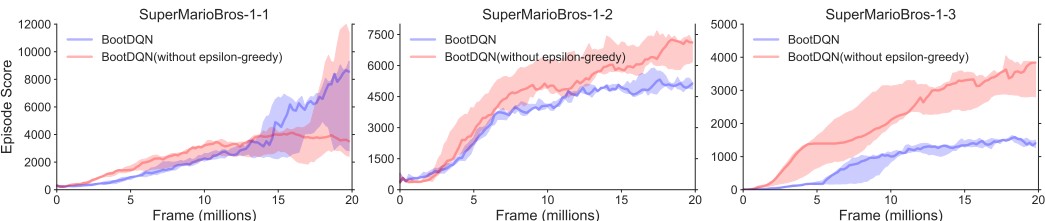

Figure 19: Comparison of BootDQN with and without $\epsilon$-greedy on SuperMarioBros.

# E  DISCUSSION

## E.1  DIRECT EXTENSION OF RLSVI COULD FAIL

In this part, we provide numerical evidence that the direct extension of RLSVI could fail for Atari tasks. In particular, we use a randomly initialized convolutional neural network as the fixed feature extractor and implement RLSVI according to Appendix A.1. Since we work with infinite horizon MDPs, we replace the non-discounted target in (A.3) by the corresponding discounted target. We display experiment results in Figure 20. In particular, we observe that such an implementation of RLSVI is unable to solve Atari tasks.

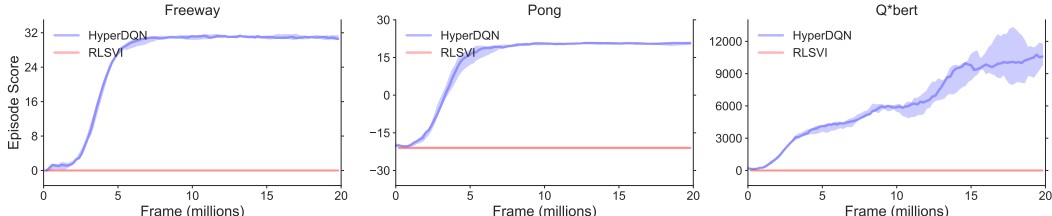

Figure 20: Comparison of RLSVI and HyperDQN on Atari.

## E.2  TRAINABILITY ISSUES OF HYPERDQN

In this part, we continue the discussion of the architecture design in Section 4.1. In particular, we illustrate why the original architecture in (Dwaracherla et al., 2020) is not suitable for RL tasks.

Table 6: Information about the base model used for deep sea.

|  | Shape ($f_{in}, f_{out}$) | Expected Magnitude of Initialization ($1/\sqrt{f_{in}}$) |
|---|---|---|
| Layer 1 | (900, 64) | 1/30 |
| Layer 2 | (64, 64) | 1/8 |
| Layer 3 | (64, 2) | 1/8 |

As we have argued in Section 4.1, the main challenge of applying the architecture in (Dwaracherla et al., 2020) is parameter initialization. To illustrate this issue, consider that we use a two-layer MLP neural network with a width of 64 for the deep sea task of size 30. Suppose we use the default PyTorch initialization[12] (i.e., sampling from the uniform distribution $U[-1/\sqrt{d_{i-1}}, 1/\sqrt{d_{i-1}}]$), which is also known as "$1/\sqrt{f_{in}}$" with $f_{in}$ being the input dimension of the $i$-th layer. We list the parameter information in Table 6 (recall that the state dimension is 900 and the action dimension is 2 for the deep sea task). We see that the expected magnitude of the parameter differs over layers. In particular, the desired magnitude is quite small compared with the magnitude of the output of the hypermodel ($\approx 1$). As a result, if we directly apply the hypermodel for all layers of the base model, the input signal over layers explodes and the gradient is quite large.

- After about 10 iterations, we observe that the parameter diverges if we use the SGD algorithm and apply the hypermodel for all layers of the base model.
- Even if we use adaptive algorithms like Adam, the training result is not expected; see the empirical result in Figure 21. In particular, if we apply the hypermodel for all layers of the base model, $Q$-values are very large and the resulting algorithm does not succeed.

Furthermore, this parameter initialization issue becomes more severe when we use deep convolution neural networks for Atari and SuperMarioBros tasks.

## E.3  $\epsilon$-GREEDY IN MANY EFFICIENT ALGORITHMS

We realize many efficient algorithms (Osband et al., 2016a; Rashid et al., 2020; Bai et al., 2021) still use $\epsilon$-greedy in practice. However, we argue that $\epsilon$-greedy is not efficient since it cannot write

---

[12]https://pytorch.org/docs/stable/generated/torch.nn.Linear.html

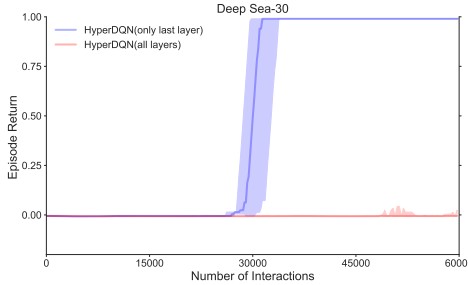
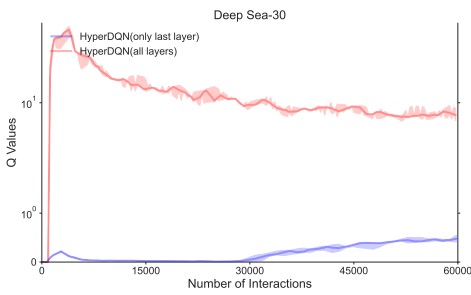

(a) Episode return on the deep sea.

(b) $Q$-values on the deep sea.

Figure 21: Comparison of HyperDQN with two configurations: 1) the hypermodel is applied only at the last layer of the base model (ours); 2) the hypermodel is applied for all layers of the base model.

off sub-optimal actions after experimentation. In addition, combining with $\epsilon$-greedy violates the principle of commitment as discussed in Section 4.5. We conjecture existing practical algorithms combine their exploration strategies with $\epsilon$-greedy mainly due to imperfect imitation of theoretical algorithms, which is analyzed below. This direction deserves further investigation.

On the one hand, we believe BootDQN (Osband et al., 2016a) uses $\epsilon$-greedy is to improve the diversity since finite ensembles could offer limited diverse action sequences. For instance, if the number of actions is larger than the number of ensembles, BootDQN cannot produce all possible action sequences to explore at the initial stage. In this case, $\epsilon$-greedy is somehow helpful. Another explanation is that BootDQN just follows the default setting of DQN (Mnih et al., 2015). In Figure 19, we see that the variant that drops the $\epsilon$-greedy could be better than the original implementation of BootDQN in some tasks.

On the other hand, there are many practical issues to implement OFU-based algorithms in real applications, as discussed in Section 3. In fact, directly adding an exploration bonus yields unexpected outcomes. Let us (re-)illustrate this using the example of the deep sea (shown in Figure 5). Imagine the agent visits a certain path to collect the data; for instance, the red path in Figure 22. With the experience replay buffer, the agent can only update the $Q$-value function with state-action pairs from the visited path. By exploration bonus, such state-action sequences would have larger $Q$-values. During the next episode, the agent would repeat the same action sequences by following the greedy policy. It turns out that theoretical algorithms like (Jin et al., 2018) do not have such an issue. This is because theoretically, algorithms in (Jin et al., 2018) can use an optimistic initialization. With the optimistic initialization, the algorithm instead would select other actions rather than visited actions. In practice, however, it is not easy to obtain an optimistic initialization with deep neural networks (Rashid et al., 2020). Again, $\epsilon$-greedy could somehow avoid this issue.

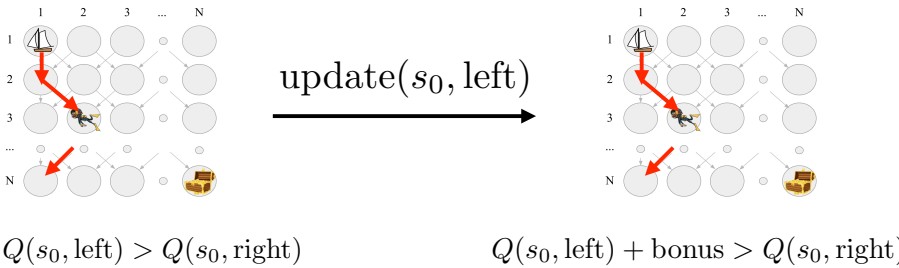

Figure 22: Illustration for the initialization issue of OFU-based algorithms. After the initialization, action "left" dominates at state $s_0$. After the experience replay buffer, action "left" dominates again at state $s_0$. This issue is caused by the pessimistic initialization by neural network (Rashid et al., 2020).

In addition to the initialization issue, the uncertainty (i.e., the exploration bonus) should propagate over stages in a backward manner. In (Jin et al., 2018), this operation is implemented by the exact dynamic programming. In contrast, practical algorithms randomly draw mini-batch from the

experience buffer to update, which may not properly propagate the uncertainty. As a result, the $Q$-values are not always optimistic, and induced action sequences may not be diverse. This issue is pointed out in (Bai et al., 2021). Again, $\epsilon$-greedy may be helpful to address this issue.

### E.4 Discussion of NoisyNet

In this part, we explain the differences between HyperDQN and NoisyNet in detail. The following discussion aims to provide insights about *what HyperDQN can achieve while NoisyNet cannot*. Note that we are by no means criticizing NoisyNet. Instead, NoisyNet is simple and has strong empirical performance. We hope the above discussion could provide intuitions (or explanations) why NoisyNet succeeds or fails under different cases.

- First, as remarked in (Fortunato et al., 2018), NoisyNet is not ensured to approximate the posterior distribution of parameters. Therefore, NoisyNet is not a typical Thompson sampling based algorithm. The implication is that we may not use the well-known theory (Osband et al., 2013; 2019; Zanette et al., 2020) to analyze NoisyNet.
- Second, NoisyNet re-samples a new policy every time step (Line 5 of Algorithm 1 in (Fortunato et al., 2018)). Consequently, it does not implement deep exploration like RLSVI (see (Osband, 2016b, Section 4.1)). This may explain the empirical result that NoisyNet can not solve the deep sea task when the problem size is large than 20 (see Figure (Osband et al., 2018, Figure 9)). Instead, BootDQN and HyperDQN can solve the deep sea task even if the problem size is large than 20.
- Third, NoisyNet does not have the mechanism of "prior" (Osband et al., 2018) even though it is randomized. As a consequence, NoisyNet cannot leverage an informative prior to accelerate exploration when such information is available. In contrast, HyperDQN can achieve this goal as discussed in Appendix E.6.

### E.5 Extension to Continuous Control

In this part, we briefly discuss how to extend the idea in this paper for continuous control tasks. We also present some preliminary results to support this direction.

Following the idea presented in Section 4.3, we should leverage the hypermodel to capture the posterior distribution of the $Q$-value function. Considering the standard actor-critic methods, we can replace the vanilla critic with the one that is built by the hypermodel. In particular, we replace the last layer of the critic with a hypermodel. In this way, each $z$ corresponds to a specific critic from the posterior distribution. To perform policy optimization, we notice that each actor (i.e., a policy network) should be greedy to a specific critic with the same index $z$. To this end, the last layer of the actor network is also built by a hypermodel. The architecture design is illustrated in Figure 23.

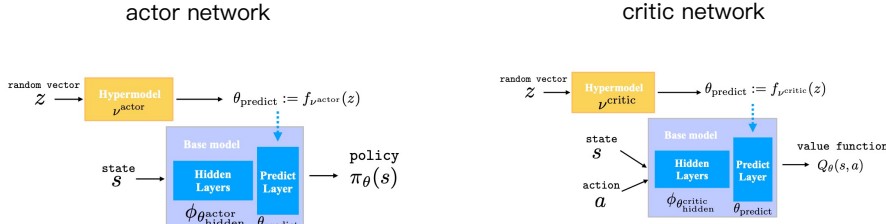

Figure 23: Illustration for HAC. In addition to a hypermodel in the critic network, there is also a hypermodel for the actor network.

Let $(\theta_{\text{hidden}}^{\text{critic}}, \nu^{\text{critic}})$ be the parameter for the critic network and $(\theta_{\text{hidden}}^{\text{actor}}, \nu^{\text{actor}})$ be the parameter for the actor network. Following the optimization framework in (Lillicrap et al., 2016; Haarnoja et al., 2018), the training objective for the actor network is:

$$\max_{\theta_{\text{hidden}}^{\text{actor}}, \nu^{\text{actor}}} \sum_{z \in \widetilde{\mathcal{Z}}} \sum_{s \in \widetilde{\mathcal{D}}} Q_{\theta_{\text{hidden}}^{\text{critic}}, \nu^{\text{critic}}}(s, a_z, z), \quad \text{with } a_z \sim \pi_{\theta_{\text{hidden}}^{\text{actor}}, \nu^{\text{actor}}}(s; z), \tag{E.1}$$

---

**Algorithm 3** HyperActorCritic(HAC)

---

**for** episode $k = 0, 1, 2, \cdots$ **do**
    generate a random vector $z \sim \mathcal{N}(0, I)$.                                          ▷ Sampling
    instantiate an actor $\pi_\theta(\cdot; z)$.
    **for** stage $t = 0, 1, 2, \cdots, T-1$ **do**                                 ▷ Interaction
        observe state $s_t$.
        sample the action $a_t \sim \pi_\theta(s_t; z)$.
        receive the next state $s_{t+1}$ and reward $r(s_t, a_t)$.
        sample $\xi$ uniformly from unit hypersphere.
        store $(s_t, a_t, r, \xi, s_{t+1})$ into the replay buffer $\mathcal{D}$.
        agent step $n \leftarrow n + 1$.
        **if** `mod` (agent step $n$, train frequency $M$) $== 0$ **then**         ▷ Update
            sample a mini-batch $\widetilde{\mathcal{D}}$ of $(s, a, r, \xi, s')$ from the replay buffer $\mathcal{D}$.
            sample $N$ random vectors $z_i'$: $\widetilde{\mathcal{Z}} = \{z_i'\}_{i=1}^N$.
            optimize the critic network using the loss function (E.2) with $\widetilde{\mathcal{D}}$ and $\widetilde{\mathcal{Z}}$.
            optimize the actor network using the loss function (E.1) with $\widetilde{\mathcal{D}}$ and $\widetilde{\mathcal{Z}}$.
            update the critic target network with exponential moving average.
        **end if**
    **end for**
**end for**

---

where $Q_{\theta_{\text{hidden}}^{\text{critic}}, \nu^{\text{critic}}}$ is the critic network and $\pi_{\theta_{\text{hidden}}^{\text{actor}}, \nu^{\text{actor}}}$ is the actor network. In particular, objective (E.1) states that we should sample actions from the actor network to maximize the $Q$-value function. The difference with the traditional actor-critic methods is that we sample many actor networks (indexed by $z$) to optimize.

Similarly, the training objective for the critic network is:

$$\min_{\theta_{\text{hidden}}^{\text{critic}}, \nu^{\text{critic}}} \sum_{z \in \widetilde{\mathcal{Z}}} \sum_{(s,a,r,s',\xi) \in \widetilde{\mathcal{D}}} \left( Q_{\theta_{\text{hidden}}^{\text{critic}}, \nu^{\text{critic}}}(s, a, z) - \sigma_w \xi^\top z - \left( r + \gamma Q_{\bar{\theta}_{\text{hidden}}^{\text{critic}}, \bar{\nu}^{\text{critic}}}(s', a_z', z) \right) \right)^2,$$

(E.2)

with $a_z' \sim \pi_{\theta_{\text{hidden}}^{\text{actor}}, \nu^{\text{actor}}}(s', z)$. In (E.2), $Q_{\bar{\theta}_{\text{hidden}}^{\text{critic}}, \bar{\nu}^{\text{critic}}}$ is the target network. The goal of (E.2) is to perform the temporal difference learning to optimize the critic. Note that the prior regularization term is not appeared in (E.2) for easy presentation.

We call the above method as HyperActorCritic(HAC). Its implementation is outlined in Algorithm 3, which shares many features with Algorithm 2.

Now, we consider the Cart Pole task from `dm_control`[13] (Tunyasuvunakool et al., 2020) as a testbed; see Figure 24a . Detailed environment information is provided as follows: the dimension of the state space is 5, the dimension of the action space is 1, the planning horizon is 1000 and the reward is between 0 and 1. This environment is a standard platform to test the exploration efficiency for continuous control algorithms because the agent can only obtain $+1$ reward when it succeeds and obtain 0 reward otherwise.

We compare our extension (HAC) with the strong baseline SAC (Haarnoja et al., 2018). Even though the policy entropy reward is used to guide exploration, SAC does not have the epistemic uncertainty qualification like HAC. As a result, we expect SAC does not perform well for Cart Pole. The numerical result is displayed in Figure 24b. We see that HAC outperforms SAC on this hard exploration task.

### E.6 WHEN AN INFORMATIVE PRIOR IS AVAILABLE

In this part, we briefly discuss the role of prior value functions in our formulation. In particular, we argue that if an informative prior value function is available, the exploration efficiency can be significantly improved.

---

[13]https://github.com/deepmind/dm_control

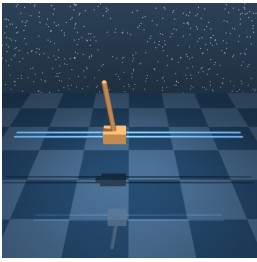

(a) Illustration for Cart Pole.

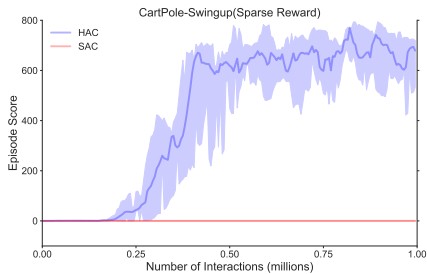

(b) Episode return on Cart Pole.

Figure 24: Comparison of HAC and SAC on the hard exploration Cart Pole (Tunyasuvunakool et al., 2020).

Here we show the learning curve with an informative prior in Figure 25. In particular, this informative prior value function is obtained from the pre-trained model after 20M frames. We see that this informative prior value function improves efficiency a lot. Note that the main goal of this experiment is to illustrate that an informative prior could accelerate exploration. We will consider how to automatically acquire an informative prior with human demonstrations (Hester et al., 2018; Sun et al., 2018) in the future.

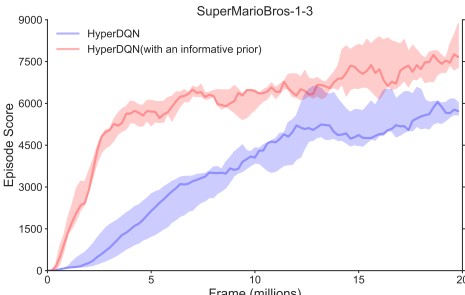

Figure 25: Comparison of HyperDQN with and without an informative prior on SuperMarioBros-1-3.

