# OpenReview forum: "HyperDQN: A Randomized Exploration Method for Deep Reinforcement Learning"
_ICLR.cc/2022/Conference — ICLR 2022 Poster_

### Official Review · Reviewer_2epQ · 2021-10-31

**Correctness:** 3
**Technical Novelty And Significance:** 2
**Empirical Novelty And Significance:** 3
**Recommendation:** 6
**Confidence:** 3

**Main Review:**

### Strengths
Overall, I think it is a good trial to extend the idea of Hypermodel to the realm of deep reinforcement learning. Compared with BootDQN, *HyperDQN* will require much less computational power. Meanwhile, the paper provides a large amount of experiment results to show the superior performance of *HyperDQN*. Meanwhile, the paper is also written in a clear logical flow.

### Weaknesses
I apologize if I understand incorrectly. One of my major concern is on the novelty. Given the existence of Hypermodel, the *HyperDQN* seems to be a very straightforward application of it on deep reinforcement learning without more insights. Meanwhile, from my perspective, the Theorem 1 simply re-writes the motivation of Hypermodel in a mathematical way, using linear hypermodel to approximate the posterior distribution, which is already explained in [1].

Another concern I have is about the performance of *HyperDQN*. Specifically, although the experiments compare *HyperDQN* with BootDQN, it seems that this paper uses the version of BootDQN based on [2]. However, the version of BootDQN in [3] has much better performance than that in [2]. Further, it seems the BootDQN in [3] did not use $\epsilon$-greedy. Have you compred *HyperDQN* with the BootDQN in [3]?

**Honestly, I'm not very experienced in running experiments, so I apologize if the above comments on experiments contain some flaws.**

### Questions
- Does *HyperDQN* contain other insights that are not presented in [1]?
- Did the comparison experiments use BootDQN in [2] or [3]?
- Did you compare *HyperDQN* and BootDQN on deep sea?
- When you ran BootDQN, did you use number of ensembles significantly less than the one used in [3]?

### Suggestions on Writing
- In appendix C, it seems $b$ should be replaced by $\nu_b$ in equation C.6.
- At the bottom of page 4, it says $\theta_{\mathrm{predict}}\in\mathbb{R}^{d\times N_a}$, but in the previous paragraph, it also says $\theta_{\mathrm{predict}}\in\mathbb{R}^d$. This can potentially be confusing.
- In Theorem 1, it is may be better to replace "then $\theta:=\dots$ with" by "then $\theta:=\dots$ satisfies"

```
[1] Vikranth Dwaracherla, Xiuyuan Lu, Morteza Ibrahimi, Ian Osband, Zheng Wen, and Benjamin Van Roy. Hypermodels for exploration. In Proceedings of the 8th International Conference on Learning Representations, 2020.

[2] Ian Osband, Charles Blundell, Alexander Pritzel, and Benjamin Van Roy. Deep exploration via bootstrapped DQN. In Advances in Neural Information Processing Systems 29, pp. 4026–4034, 2016a.

[3] Ian Osband, John Aslanides, and Albin Cassirer. Randomized prior functions for deep reinforcement learning. In Advances in Neural Information Processing Systems 31, pp. 8626–8638, 2018.
```

**Summary Of The Paper:**

This paper proposes a new posterior sampling-based exploration method, called *HyperDQN*, for deep reinforcement learning. Conceptually, *HyperDQN* is modified based on randomized least-square value iteration (RLSVI) and it resolves the limitations of RLSVI on feature engineering and high computational complexity. Technically, *HyperDQN* extends the Hypermodel technique, which is only used for bandit learning in its original paper. The performance of *HyperDQN* is evaluated on Atari 2600 games, SuperMarioBros and deep sea environment.

**Summary Of The Review:**

The paper proposes a computationally lighter method *HperDQN* for exploration in deep reinforcement learning and shows its superior performance in experiments. However, the formulation of *HyperDQN* is conceptually not novel and its performance compared to other methods is also questionable.

---

> ### Author Response · Authors · 2021-11-16
> **Response to Reviewer 2epQ (Part 1/2)**
>
> We thank you for your valuable review.
>
> **Comment 1**:  Theorem 1 simply re-writes the motivation of Hypermodel, using linear hypermodel to approximate the posterior distribution, which is already explained in [1] in a mathematical way.
>
> **Answer 1**: Thanks for your comments. We kindly point out that our Theorem 1 is not just a ‘’re-write’’ on the motivation of hypermodel. We also delivered two important messages that are not conveyed  in [1].
>
> 1.  We proved  ''the hypermodel can approximate the posterior distribution **under objective (4.1)**'', a fundamental question that is NOT answered in Dwaracherla et al. (2020). Building the connection between ‘’hypermodel’’ and ''objective (4.1)’’ is important:  without Theorem 1, the objective (4.1) will remain as a pure heuristic with no theoretical guarantee.
>
> Let us emphasize the difference. As mentioned by the reviewer, [1] explained ‘linear hypermodel can approximate the posterior distribution’. However, they only show that ‘’there  **exists** such a hypermodel to approximate posterior’’, but they did not tell us how to find it. This question is explored in our Theorem 1.
>
> 2. We also deliver a message that the **$z$-dependent noise** is important in objective (4.1). This message is not explained or validated in (Dwaracherla et al., 2020) since their theorem is unrelated to objective (4.1). Without our Theorem 1, readers may believe that an independent Gaussian noise $\omega$ would work well in objective (4.1) and achieve the goal of posterior approximation. However, we prove that this is not true. See the explanation in Remark 1, the analysis in Appendix C.4, and the empirical evidence in Figure 2.
>
>
> **Question 2**:  ''the HyperDQN seems to be a very straightforward application of it on deep reinforcement learning without more insights.'' ''Does HyperDQN contain other insights that are not presented in [1]?''
>
> **Answer 2**: We kindly point out that HyperDQN is NOT a straightforward extension. Importantly, there are new insights in our paper that are not presented in [1], both theoretically and empirically. We explain as follows.
>
> **Theoretical insight**:  We have discussed in  **Answer 1**.
>
> **Empirical insight**: It is non-trivial to **jointly optimize the feature extractor and posterior samples**, which is not covered in [1]. In fact, a direct application of the original hypermodel in (Dwaracherla et al., 2020) for deep RL tasks will FAIL; see the evidence in Appendix F.2. Importantly, our architecture is different from the one used in (Dwaracherla et al., 2020). Specifically, Dwaracherla et al. (2020) apply the hypermodel for **all** layers of the base model to solve simple bandit tasks (refer to (Dwaracherla et al., 2020, Figure 1)). On the other hand, we extend the hypermodel to the deep RL case by applying the hypermodel for the **last** layer of the value function.
>
> The modification (all layers v.s. the last layer) is motivated by the **trainability** issue. In particular, **the direct extension of (Dwaracherla et al., 2020) would fail under the deep RL scenario**. The underlying reason is the **parameter initialization** issue. Concretely, the output of the initialized hypermodel is not a good initialization for the parameter of the base model when we use the architecture in (Dwaracherla et al., 2020). In particular, modern initialization techniques (e.g., LeCun's initialization) suggest we should initialize the parameter of the $i$-th layer by sampling from the Gaussian distribution $\mathcal{N}(0, \sigma^2)$ with $\sigma=1/\sqrt{d_{i-1}}$, where $d_{i-1}$ is the width of the $(i-1)$-th layer.However, the architecture in (Dwaracherla et al., 2020) cannot achieve this. Instead, the architecture in (Dwaracherla et al., 2020) implies the magnitude of the parameter of the base model is in [-1, 1]. As a result, the input signal amplifies over layers and the gradient explodes when we use the architecture in (Dwaracherla et al., 2020).
>
> In fact, **Dwaracherla et al. (2020) only use a two-layer base model with a width of 10 for bandit tasks**, so the trainability issue is not severe in their applications. But **for deep RL, we use larger neural networks with complicated architectures including CNN.** Thus, this trainability issue is severe, which motivates us to use the simple architecture in our paper. In our model, the hidden layers of the base model are initialized with common techniques and the last layer could be properly initialized by normalizing the output of the hypermodel. This addresses the parameter initialization issue. Fortunately, this simple architecture still retains the main ingredient of RLSVI (i.e., capturing the posterior distribution over a linear prediction function).
>
> In summary, HyperDQN is NOT just a straightforward extension of [1]. New important modifications and insights are provided.

---

> > ### Author Response · Authors · 2021-11-17
> > **We have revised our Answer 1**
> >
> > We notice that the reviewer has revised his/her claim regarding Comment 1 in the review. Therefore, we update our answer accordingly. Please re-check our revised response in **Answer 1**.
> >
> > Thanks for your time and consideration.

---

> > ### Comment · Reviewer_2epQ · 2021-11-30
> > **Score Increase**
> >
> > Thank you very much for your detailed reply. All my concerns about experiments are well-addressed.
> >
> > I don't think Theorem 1 to be novel mainly because I believe its major novelty lies in the objective formulation and using $z$-dependent noise, both of which were proposed in Dwaracherla et al. (2020) as you also mentioned. Meanwhile, the results of Theorem 1 looks more like a straightforward calculation obtained by plugging in standard linear model assumption.
> >
> > However, even though I'm not very satisfied with the theoretical novelty, I appreciate the extensive experiments (together with those after the rebuttal) that have been done and the promising performance of HyperDQN. As a result, I think this work is still worth for a publication and thus I have increased my score.

---

> ### Author Response · Authors · 2021-11-16
> **Response to Reviewer 2epQ (Part 2/2)**
>
> **Question 3**: Did the comparison experiments use BootDQN in [2] or [3]?
>
> **Answer 3**: We use the version in [3]. That is, the version with prior value functions. It is not clear whether [3] uses epsilon-greedy or not. Since the main ingredient of [3] is the prior value function, we keep the epsilon-greedy design. This configuration is also used in [5] (please also refer to the famous implementation at GitHub: https://github.com/johannah/bootstrap_dqn ). Indeed, we have provided the results of BootDQN without epsilon-greedy in the submission, which corresponds to Figure 14 in the revision. We see that without epsilon-greedy, BootDQN becomes better on SuperMarioBros-1-2 and SuperMarioBros-1-3 but becomes worse on SuperMarioBros-1-1. The improved results are still inferior to HyperDQN.
>
>
> ---
>
> **Question 4**: Did you compare HyperDQN and BootDQN on deep sea?
>
> **Answer 4**: In Table 4 in Appendix E.3, we have shown that HyperDQN is much efficient than BootDQN.
>
> ---
>
> **Question 5**: When you ran BootDQN, did you use number of ensembles significantly less than the one used in [3]?
>
> **Answer 5**: Yes. In this paper, we implement BootDQN with 10 ensembles, which follows the conﬁguration in Section 6.1 of [2]. However, in [3], BootDQN is implemented with 20 ensembles. We remark that the choice of 10 ensembles is commonly used in the previous literature [4, 5] since it is more computationally cheap.
>
> To address your concern, we provide the ablation study about this choice in Figure 8 in Appendix D.5. Unfortunately, we do not see significant gains when using 20 ensembles.
>
> ---
>
> **Comment 6**: Writing suggestions.
>
> **Answer 6**: Thanks for your suggestions. We have revised the related parts to make them clear.
>
> ---
>
> [4] Rashid, Tabish, et al. "Optimistic exploration even with a pessimistic initialisation." ICLR, 2020.
>
> [5] Bai, Chenjia, et al. "Principled exploration via optimistic bootstrapping and backward induction." ICML, 2021.

---

### Official Review · Reviewer_C9G4 · 2021-11-01

**Correctness:** 4
**Technical Novelty And Significance:** 3
**Empirical Novelty And Significance:** 3
**Recommendation:** 8
**Confidence:** 4

**Main Review:**

Strengths:
+ The outcome of the paper which adapts the ideas from RLSVI to deep reinforcement learning settings is significant and practical.
+ The paper is very well written, clear, and easy to understand. The supplementary materials provide most of the information needed for acquiring a better understanding of the paper.
+ The authors provide theoretical analysis and proofs to support their main ideas and design choices.
+ The experimental results show superior performance of HyperDQN compared to state-of-the-art algorithms.
+ The results are reproducible as the implementation and experimental details are provided in the paper.

Weaknesses:
- The proposed approach provides incremental contributions w.r.t. the existing ideas from the previous work.
- The contributions are very specific to a class of problems and algorithms.

Comments:
- The main ideas of the paper has been previously introduced in the literature and is further extended to the deep RL scenarios which make the presented contributions only somewhat novel.
- The paper can be improved by providing some information and\or intuition about the generalizability of the proposed approach to the other domains. Additionally, discussions about the challenges in extending the proposed idea to the areas such as continuous control, offline RL, etc. would be highly appreciated.
- Could you elaborate about the challenges and potential problems that could arise when jointly optimizing the hypermodel and the feature extractor? One of the main contributions of the paper is addressing the major problems with RLSVI which are restricting its applications in deep RL. While addressing these limitations regarding computational costs of RLSVI is discussed extensively, providing a specific cost function to jointly optimize the feature extractor and the hypermodel might lead to stability and convergence problems. Hence, it would be beneficial if the authors provide further insights into this matter.


**Summary Of The Paper:**

The paper introduces HyperDQN, a novel exploration technique for deep reinforcement learning, which extends the ideas from an existing method, namely RLSVI, to the deep RL settings. HyperDQN optimizes a linear hypermodel, which is used to compute state-action values as a function of extracted features, to work as a module that converges to a sample from the posterior distribution of the Q-value function. The authors further present a specific cost function to jointly optimize the hypermodel and the feature extractor that can efficiently address the limitation of the RLSVI approach in deep RL scenarios. Empirical study demonstrates significant improvement compared to the competitors on several benchmarks.

**Summary Of The Review:**

Overall, the paper introduces a novel approach for randomized exploration in deep RL, is well written, and shows strong promise in terms of performance compared to other state-of-the-art methods. Additionally, it includes valuable theoretical and experimental discussions that provide further support for their contributions. I thus vote for an accept.

---

> ### Author Response · Authors · 2021-11-16
> **Response to Reviewer C9G4  (Part 1/2)**
>
> We highly appreciate your positive opinion about valuable theoretical results and experimental discussions.
>
> **Comment 1**: The proposed approach provides incremental contributions w.r.t. the existing ideas from the previous work.
>
> **Answer 1**: Thanks for your comment. We would like to clarify that it is not trivial to extend the hypermodel to deep RL. In particular, the trainability issue matters for deep RL. See more detailed discussion in **Answer 3**.
>
> **Comment 2**: ''The contributions are very specific to a class of problems and algorithms.''
> ''The paper can be improved by providing some information and\or intuition about the generalizability of the proposed approach to the other domains.''
>
> **Answer 2**: Thanks for your comment. As you noticed, HyperDQN is tailored to the problem of Deep RL, and it is specific to improving randomized exploration algorithms.
> Nevertheless, we would like to point out that our idea is not restricted to  (online) Deep RL, we briefly discuss the extension to offline RL, continuous control task as follows.
>
> For offline RL, a direct extension is to replace finite ensembles or drop out in existing methods (Yu et al., 2020; Wu et al., 2021) with hypermodel since the hypermodel is effective to capture the epistemic uncertainty. This topic will be considered in the future work.
>
> For continuous control tasks, we could leverage the actor-critic method to design an efficient exploration strategy. A natural extension is to replace the last layer of a critic network with the hypermodel, which allows us to approximate the posterior distribution of the $Q$-value function; see the architecture design in Figure 24 in Appendix F.5. The preliminary result on the hard exploration task Cart Pole is shown in Figure 25 in Appendix F.5. In particular, SAC can not solve this toy problem while our extension succeeds because the epistemic uncertainty measure in the hypermodel leads to efficient exploration.

---

> ### Author Response · Authors · 2021-11-17
> **Response to Reviewer C9G4 (Part 2/2)**
>
>
> **Question 3**: Could you elaborate about the challenges and potential problems that could arise when jointly optimizing the hypermodel and the feature extractor? The specific cost function to jointly optimize the feature extractor and the hypermodel might lead to stability and convergence problems.
>
> **Answer 3**: Thanks for pointing this out. It is true that jointly optimization hypermodel+feature extractor is challenging. However, we successfully tackled the training stability and convergence issue by introducing the following techniques and tricks.
>
> 1. Stability issue. It is known that parameter initialization contributes a lot to ensuring training stability.   In Deep RL, we observe that the original model in  (Dwaracherla et al., 2020) suffers severe training instability (the gradient explodes at the initialization). We further find out that inappropriate initialization is the main reason to blame.  We addressed this issue using a modified architecture.  We explain as follows.
>
> Dwaracherla et al. (2020) apply the hypermodel for **all** layers of the base model to solve simple bandit tasks (refer to (Dwaracherla et al., 2020, Figure 1)). In contrast, we extend the hypermodel to the deep RL case by applying the hypermodel for the **last** layer of the value function.
>
> The modification (all layers v.s. the last layer) is due to the training stability issue. Concretely, the output of the initialized hypermodel is not a good initialization for the parameter of the base model when we use the architecture in (Dwaracherla et al., 2020). In particular, modern initialization techniques (such as LeCun's initialization) suggest we should initialize the parameter of the $i$-th layer by sampling from the Gaussian distribution $\mathcal{N}(0, \sigma^2)$ with $\sigma=1/\sqrt{d_{i-1}}$, where $d_{i-1}$ is the width of the $(i-1)$-th layer.  However, the architecture in (Dwaracherla et al., 2020) cannot achieve this. Instead, the architecture in (Dwaracherla et al., 2020) implies the magnitude of the parameter of the base model is in [-1, 1]. As a result, the input signal amplifies over layers and the gradient explodes.
>
> In fact, **Dwaracherla et al. (2020) only use a two-layer base model with a width of 10 for bandit tasks**, so the stability issue is not severe in their applications. But **for deep RL, we use larger neural networks with complicated architectures including CNN.** Thus, this training stability issue is severe, which motivates us to use the simple architecture in our paper. In our model, the hidden layers of the base model are initialized with common techniques and the last layer could be properly initialized by normalizing the output of the hypermodel. This addresses the parameter initialization issue, i.e. avoiding the initial gradient explosion. Fortunately, this simple architecture still retains the main ingredient of RLSVI (i.e., capturing the posterior distribution over a linear prediction function).
>
>
>
> 2. Convergence issue. In practice, the algorithmic convergence relies on the following factors: 1. initialization, 2. TD convergence.
>  We have discussed initialization in the stability part. As for TD convergence, we adopt the current tricks such as target networks, experience replay.  Putting all these ingredients together, we did not observe the case of divergence in our experiments.
>
>
> We have clarified these points in Section 4.2 and Appendix F.2 in the revision. We hope this answer can address your concern.
>
>
> ---
>
> Wu, Yue, et al. "Uncertainty Weighted Actor-Critic for Offline Reinforcement Learning." ICML 2021.
>
> Yu, Tianhe, et al. "Mopo: Model-based offline policy optimization." NeurIPS 2020.
>
> Dwaracherla, Vikranth, et al. "Hypermodels for exploration." ICLR 2020.

---

### Official Review · Reviewer_TPoj · 2021-11-02

**Correctness:** 3
**Technical Novelty And Significance:** 2
**Empirical Novelty And Significance:** 3
**Recommendation:** 3
**Confidence:** 4

**Details Of Ethics Concerns:**

This paper is about deep reinforcement learning problems on simulated domains. Therefore there is no ethics issue associated.

**Main Review:**

$\textbf{Strength}$:
- This work tackles an important problem of exploiting randomized exploration methods in deep reinforcement learning problems.
- This paper is very well written and easy to follow.
- Overall the method of combining hypermodel with Q-network is quite sound and very general to be applied on deep RL problems.
- The empirical evaluation results are very extensive where relevant baselines are considered for comparison.

$\textbf{Weakness}$:
- Very limited novelty where the idea is from the work [1].
- During comparison, only results at intermediate training stage  (i.e., 20M frames)  is presented while the detailed scores for the full training is missing. So it is unclear if the method could really be the next state-of-the-art method for Atari 2600 test suite.
-Only limited number of baselines are considered. There is not decent exploration methods, such as UCB-type exploration or ramdomized exploration with noisy nets, being considered for comparison.


$\textbf{Detailed comments}$:

1. My primary concern is about the novelty of this paper. I found that both the formulation for hypermodel and the loss function for optimizing the hypermodel is inherited from the existing work [1]. In [1], a more general formulation for the hypermodel has been discussed which includes linear models, neural models and additive prior models. In HyperDQN, a linear hypermodel is adopted where there is nothing new for the method. Thus the contribution of this paper is not in terms of introducing a new method but is to show additional experimental results for an existing method on several deep RL domains. I feel such contribution might not be significant enough to be published as a full paper in ICLR.

2. This paper tackles the policy learning problem from a direction of considering randomized exploration method. It claims the randomized exploration method O2BI as a SOTA exploration method but I do not think so. Generally, the O2BI as well as HyperDQN is only compared with limited baselines when the training is incomplete. I feel that many other variants of UCB-type algorithms, such as noisy net [2] might be able to outperform O2BI or HyperDQN greatly with 200M training budget. For instance, prediction-error based exploration methods could effectively progress on the extremely hard exploration task Montezuma's Revenge while O2BI and HyperDQN make no progress on it. Therefore, a more inclusive exploration methods for discussion/comparison is desired.

3. The training curves presented in Fig 12 come with no error bars and/or O2BI is missing from the curves.

4. I feel noisy net is also a related randomized exploration model which should be considered for discussion and/or experimental comparison.

[1] HYPERMODELS FOR EXPLORATION
[2] NOISY NETWORKS FOR EXPLORATION

**Summary Of The Paper:**

This paper proposes a new randomized exploration method termed HyperDQN.

It aims to improve the popular Randomized Least Square Value Iteration (RLSVI) method by addressing RLSVI's major limitations of computational burden and its requirement of known good features in advance to adopt the linear setting.

The principal of HyperDQN is to combine DQN with a probabilistic hypermodel which could generate posterior samples for the parameter values of the weights at the last linear layer of the Q-function. Both the hypermodel and the Q-function could be jointly optimized during training. By sampling the Q-function parameters, it helps the model to select exploratory action sequences.

 Empirical evaluation results are provided in Atari 2600 domain, SuperMarioBros and deep sea. Overall, the proposed method could outperform Bootstrapped DQN, Double DQN and another recent bootstrapping baseline termed OB2I in majority of the testified domains.

**Summary Of The Review:**

This paper is a well written one with extensive experimental results.

The novelty of this method is rather limited as the method is exactly the same as the one proposed in a prior work.

The empirical evaluation results appear to be less convincing to me because the results on Atari 2600 are only shown for intermediate model trained after 20M frames and the full training scores (inferred from the learning curves in Fig 12) appear to be much inferior than another variant of randomized exploration model which is the noisy net.

---

> ### Author Response · Authors · 2021-11-16
> **Response to Reviewer TPoj (Part 1/3)**
>
> Thanks for your valuable comments.
>
> **Comment 1**: novelty and contribution of this work compared with (Dwaracherla et al., 2020).  Both the formulation for hypermodel and the loss function for optimizing the hypermodel is inherited from the existing work [1]
>
> **Answer**: Thanks for your comments. We have to apologize that we do not clarify the architecture difference with (Dwaracherla et al., 2020) in the submission. This leads to your claim that "the method is exactly the same as the one proposed in a prior work".
> In fact,  we have changed the formulation for hypermodel. Even though the loss function is the same, we provide new insights. We elaborate as follows.
>
> (a). Change of formulation.
>
> We observe that **a direct application of the original hypermodel in (Dwaracherla et al., 2020) for deep RL tasks will FAIL**; see the empirical evidence in Appendix F.2. Importantly, our architecture **is different from the one used in (Dwaracherla et al., 2020)**. More precisely, Dwaracherla et al. (2020) apply the hypermodel for **all** layers of the base model to solve simple bandit tasks (refer to (Dwaracherla et al., 2020, Figure 1)). On the other hand, we extend the hypermodel to the deep RL case by applying the hypermodel for the **last** layer of the value function. We elaborate on this point as follows and please refer to Section 4.2 and Appendix F.2 for a detailed discussion.
>
> The modification (all layers v.s. the last layer) is essential and is motivated by the **trainability** issue. The underlying reason is the **parameter initialization** issue. Concretely, the output of the initialized hypermodel is not a good initialization for the parameter of the base model when we use the architecture in (Dwaracherla et al., 2020). In particular, modern initialization techniques (e.g., LeCun's initialization) suggest we should initialize the parameter of the $i$-th layer by sampling from the Gaussian distribution $\mathcal{N}(0, \sigma^2)$ with $\sigma=1/\sqrt{d_{i-1}}$, where $d_{i-1}$ is the width of the $(i-1)$-th layer. However, the architecture in (Dwaracherla et al., 2020) cannot achieve this. Instead, the architecture in (Dwaracherla et al., 2020) implies the magnitude of the parameter of the base model is in [-1, 1]. As a result, the input signal amplifies over layers and the gradient explodes when we use the architecture in (Dwaracherla et al., 2020); see the empirical evidence in Appendix F.2.
>
> In fact, **Dwaracherla et al. (2020) only use a two-layer base model with a width of 10 for bandit tasks**, so the trainability issue is not severe in their applications. But **for deep RL, we use larger neural networks with complicated architectures including CNN**. Hence, the trainability issue is severe, which motives us to use the simple architecture in our paper. In our model, the hidden layers of the base model are initialized with common techniques and the last layer could be properly initialized by normalizing the output of the hypermodel. This resolves the parameter initialization issue. Fortunately, this simple architecture still retains the main ingredient of RLSVI (i.e., capturing the posterior distribution over a linear prediction function).
>
> (b). New theoretical insight of the loss function of hypermodel.
>
> (b.1). We proved  ''the hypermodel can approximate the posterior distribution **under objective (4.1)**'', a fundamental question that is NOT answered in Dwaracherla et al. (2020).  Building the connection between ‘’hypermodel’’ and ‘’objective (4.1)’’ is important:  without Theorem 1, the objective (4.1) will remain as a pure heuristic with no theoretical guarantee.
>
> Different from our Theorem 1, Dwaracherla et al. (2020) prove that a linear hypermodel has sufficient representation power to approximate ANY distribution (over functions), so they emphasize it is unnecessary to use a non-linear one. However, in terms of approximating the posterior distribution (a particular distribution that we care in practice), this result only shows the **existence** of such a hypermodel,   but it does not tell us how to find it. This question is explored in our Theorem 1.
>
> (b.2) We also deliver a message that the **$z$-dependent noise** is important in objective (4.1). This message is not explained or validated in (Dwaracherla et al., 2020) since their theorem is unrelated to objective (4.1). Without our Theorem 1, readers may believe that an independent Gaussian noise $\omega$ would work well in objective (4.1) and achieve the goal of posterior approximation. However, we prove that this is not true. See the explanation in Remark 1, the analysis in Appendix C.4, and the empirical evidence in Figure 2.
>
> We hope the above answer could clarify our novelty and contribution compared with (Dwaracherla et al., 2020).

---

> ### Author Response · Authors · 2021-11-16
> **Response to Reviewer TPoj (Part 2/3)**
>
> **Comment 2(a)**: OB2I is not a SOTA exploration method.
>
> **Answer 2(a)**:  We apologize for such an imprecise argument in the submission. To address your concern, we have revised our claim in Footnote 6: “SOTA” means OB2I is a strong baseline with the 20M frames training budget as claimed in (Bai et al., 2021).
>
> **Comment 2(b)**: "UCB-type algorithms, such as noisy net might be able to outperform O2BI or HyperDQN greatly with 200M training budget".
> " the results on Atari 2600 are only shown for intermediate model trained after 20M frames and the full training scores (inferred from the learning curves in Fig 12) appear to be much inferior than another variant of randomized exploration model which is the noisy net.'"
>
> **Answer 2(b)**: Thanks for your comments. We think you make a typo that noisy net indeed is not a UCB-type algorithm.
> As for the performance after 200M,  we respectfully point out that it is not correct to "infer from the learning curves in Fig 12''. In particular, Figure 12 shows the training curves rather than the evaluation curves (that are in Figure 13 in the submission). Therefore, this inference is not reasonable.
>
> At the current stage, we cannot conduct such experiments to verify the performance of HyperDQN after 200M. The training with 200M frames suffers heavy financial burden and time cost. Specifically, running DQN for 200M frames on a single environment takes about 30 days. Unfortunately, we have limited servers so that we cannot run all experiments with 200M frames in an acceptable time. Furthermore, the experiment fee for such one experiment is about 30 * 24 * 0.5 = \\$360 as the unit cost (of the server and power) for one hour is \\$0.5. Hence, the cost for 56 environments is about \$20,160 for a single algorithm, which we cannot afford at the current stage.
>
> Nevertheless, we want to argue that the 20M frames training budget is commonly used in (Lee et al., 2019; Rashid et al. 2020, Bai et al., 2021). The main reason is that **training with 20M frames is sufficient to obtain near-optimal policies for many tasks** in Atari (e.g., Battle Zone and Pong) and SuperMarioBros (e.g., 1-1, 1-2). In this sense, we believe HyperDQN will still be a strong baseline method after 200M. (To further reach the SOTA performance after 200M, it involves more sophisticated tuning and training tricks, which will be considered as future work.)
>
> We will address your concern about the comparison with NoisyNet (after 20M) in **Answer 4**.
>
> **Comment 2(c\)**: Prediction-error based exploration methods could progress on Montezuma’s Revenge while HyperDQN make no progress on it.
>
> **Answer 2(c\)**: Thanks for your comment. It is true that prediction-error based methods outperform HyperDQN on Montezuma’s Revenge. Concretely, prediction-error based methods can achieve more than 3000 scores after 200M frames while HyperDQN is expected to obtain 0 score even after 200M frames.  The failure reason of HyperDQN is that the extremely sparse reward of Montezuma’s Revenge provides limited feedback for feature selection, which is an important factor as discussed in the introduction. In fact, *almost all randomized exploration methods like BootDQN and NoisyNet cannot perform well on this task for the same reason*: the performance of BootDQN is 500 after 200M "actor steps" (Osband et al., 2018) and the performance of NoisyNet is 3 after 200M frames (Fortunato et al., 2018). The reason why prediction-error based methods succeed on this task is explained in (Taiga et al., 2020). That is, these methods could leverage the auxiliary reward for feature selection and exploration. However, such specific architecture designs on hard exploration tasks are *unable* to generalize on other tasks in Atari as verified in (Taiga et al., 2020).
>
> Second, following the discussion, we would like to further point out that when evaluating exploration efficiency we need to consider **both** "hard" and "easy" problems for three reasons. First, we do not want algorithms to "overfit" on specific problem instances as argued in (Taiga et al., 2020). Second, before solving a task, we do not know whether this task is a "hard exploration" or "easy exploration" problem. As a result, we hope the algorithm can perform well on both types of problems. Third, even for the "easy exploration" tasks, we still need more efficient strategies than epsilon-greedy. Note that we are by no means calling for less emphasis to be placed on specific hard exploration problems. Instead, we need to pay attention to the improvement in all tasks.
>
> ---
>
> **Comment 3**: The training curves presented in Fig 12 come with no error bars and/or O2BI is missing from the curves.
>
> **Answer 3**: We have added error bars in the revision. Note that Fig 12 and 13 in the submission correspond to Fig 16 and 17 in the reversion, respectively. OB2I is not involved because we only have evaluation logs of OB2I and do not have training logs of OB2I on Atari.

---

> > ### Comment · Reviewer_2epQ · 2021-11-16
> > **A Side Comment to Montezuma’s Revenge**
> >
> > I remember that positive results about Montezuma’s Revenge were reported in the BootDQN used in [1]. Do you have any idea on why this is not reproducible from your experiments? Thank you very much!
> >
> > ```
> > [1] Ian Osband, John Aslanides, and Albin Cassirer. Randomized prior functions for deep reinforcement learning. In Advances in Neural Information Processing Systems 31, pp. 8626–8638, 2018.
> > ```

---

> > > ### Author Response · Authors · 2021-11-17
> > > **Response to the Side Comment of Reviewer 2epQ**
> > >
> > > Dear Reviewer 2epQ,
> > >
> > > Thanks for this nice observation. We explain this point in the following response.
> > > - The main reason is that the training budget in [1] is quite larger than the one used in our paper. In particular, the training budget in [1] is 800M "actor steps" (see Figure 6 in [1]) but Osband et al. do not explain the meaning of "actor steps". It seems that 1 actor step corresponds to 4 frames. In contrast, we mainly focus on the regime of 20M frames. Thus, BootDQN does not obtain positive reward for Montezuma’s revenge under the setting in our paper.
> > > - The minor reason is that Osband et al. may use a strong baseline to implement BootDQN: "the distributed DQN agent with double Q-learning, prioritized experience replay and dueling networks" (see Section 4.2.3 in [1]). Specifically, Osband et al. do not provide too many implementation details but say they follow the setting in the Ape-X [2] paper (see Appendix B.3 in [1]). In contrast, we implement BootDQN (and HyperDQN) only with double Q-learning.
> > >
> > > Hope this answer could address your concern.
> > >
> > > ---
> > >
> > > [2] Horgan, Dan, et al. "Distributed prioritized experience replay." ICLR 2018.

---

> ### Author Response · Authors · 2021-11-16
> **Response to Reviewer TPoj (Part 3/3)**
>
>
> **Comment 4**: Comparison with NoisyNet.
>
> **Answer**: Thanks for your comment. We have added NoisyNet in the related work.
>
> To compare the empirical performance, we try our best to reproduce NoisyNet and provide the numerical results in Figure 22 and Figure 23 in Appendix F.4. In particular, 6 example tasks (beam rider, montezuma's revenge, pong, seaquest, venture, and zaxxon) from the Atari suite and 6 example tasks (1-1, 1-2, 1-3, 2-1, 2-2, and 2-3) from the SuperMarioBros suite are considered with a 20M frames training budget. Reproduced results basically match the reported results in (Fortunato et al ., 2018; Figure 6). The results are summarized in the following table, which shows that HyperDQN outperforms NoisyNet on 7 out of 12 games (there is a tie on Montezuma's Revenge and SuperMarioBros-2-3). On the Deep Sea, it has been shown that NoisyNet cannot solve problems when the size is larger than 20 (Osband et al., 2018, Figure 9). In contrast, we have shown that BootDQN and HyperDQN can solve problems when the size is larger than 20 in Table 4 in Appendix E.3.
>
> |          | beam rider                | pong                     | venture                  | zaxxon                    | seaquest                 | montezuma's revenge      |
> | -------- | ------------------------- | ------------------------ | ------------------------ | ------------------------- | ------------------------ | ------------------------ |
> | NoisyNet | $\mathbf{\approx 1,700}$  | $\approx -10$            | $\approx 10$             | $\approx 1,000$           | $\mathbf{\approx 1,600}$ | $\mathbf{\approx 0}$     |
> | HyperDQN | $\approx 1,500$           | $\mathbf{\approx 21}$    | $\mathbf{\approx  300}$  | $\mathbf{\approx 4,000}$  | $\approx  600$           | $\mathbf{\approx 0}$     |
> |          | mario-1-1                 | mario-1-2                | mario-1-3                | mario-2-1                 | mario-2-2                | mario-2-3                |
> | NoisyNet | $\mathbf{\approx 12,000}$ | $\approx 6,000$          | $\approx 1,000$          | $\approx 12,500$          | $\approx 1800$           | $\mathbf{\approx 5,500}$ |
> | HyperDQN | ${\approx 9,000}$         | $\mathbf{\approx 8,000}$ | $\mathbf{\approx 5,500}$ | $\mathbf{\approx 25,000}$ | $\mathbf{\approx 2000}$  | $\mathbf{\approx 5,500}$ |
>
>
>
> In addition, we include the discussion of algorithmic differences to provide more insights on *what HyperDQN can achieve while NoisyNet cannot* in Appendix F.4. For instance, HyperDQN can achieve deep exploration while NoisyNet cannot because NoisyNet re-samples a policy at each time step. Also, there is no "prior" mechanism in NoisyNet, which is a key component for randomized exploration.
>
> We hope the above answers could address your concerns and appreciate it a lot if you can re-evaluate our paper.
>
> ---
>
> Lee, Su Young, et al. "Sample-efficient deep reinforcement learning via episodic backward update." NeurIPS 2019.
>
> Bai, Chenjia, et al. "Principled exploration via optimistic bootstrapping and backward induction." ICML, 2021.
>
> Rashid, Tabish, et al. "Optimistic exploration even with a pessimistic initialisation." ICLR 2020.
>
> Fortunato, Meire, et al. "Noisy networks for exploration." ICLR 2018.
>
> Osband, Ian, et al. "Randomized prior functions for deep reinforcement learning." NeurIPS 2018.
>
> Dwaracherla, Vikranth, et al. "Hypermodels for exploration." ICLR 2020.
>
> Taiga, Adrien Ali, et al. "On Bonus-Based Exploration Methods in the Arcade Learning Environment." ICLR,  2020.

---

> > ### Author Response · Authors · 2021-11-23
> > **We have added more experiments**
> >
> > Dear Reviewer TPoj,
> >
> > We have added additional 6 experiments: alien, amidar, battle zone, mario-3-1, mario-3-2, and mario-3-3. The final performance of learned policies on 18 benchmarks is summarized in the following table. See Figure 22 and Figure 23 in Appendix F.4 for the learning curves.
> >
> > |          | alien                     | amidar                   | battle zone               | beam rider                | pong                     | venture                  | zaxxon                    | seaquest                  | montezuma's revenge      |
> > | -------- | ------------------------- | ------------------------ | ------------------------- | ------------------------- | ------------------------ | ------------------------ | ------------------------- | ------------------------- | ------------------------ |
> > | NoisyNet | ${\approx 1,000}$         | ${\approx 200}$          | ${\approx 4,000}$         | $\mathbf{\approx 1,700}$  | $\approx -10$            | $\approx 10$             | $\approx 1,000$           | $\mathbf{\approx 1,600}$  | $\mathbf{\approx 0}$     |
> > | HyperDQN | $\mathbf{\approx 1,100}$  | $\mathbf{\approx 300}$   | $\mathbf{\approx 11,000}$ | $\approx 1,500$           | $\mathbf{\approx 21}$    | $\mathbf{\approx  300}$  | $\mathbf{\approx 4,000}$  | $\approx  600$            | $\mathbf{\approx 0}$     |
> > |          | mario-1-1                 | mario-1-2                | mario-1-3                 | mario-2-1                 | mario-2-2                | mario-2-3                | mario-3-1                 | mario-3-2                 | mario-3-3                |
> > | NoisyNet | $\mathbf{\approx 12,000}$ | $\approx 6,000$          | $\approx 1,000$           | ${\approx 12,500}$        | $\approx 1,800$          | $\mathbf{\approx 5,500}$ | $\approx 10,000$          | $\approx 32,000$          | $\mathbf{\approx 5,500}$ |
> > | HyperDQN | ${\approx 9,000}$         | $\mathbf{\approx 8,000}$ | $\mathbf{\approx 5,500}$  | $\mathbf{\approx 25,000}$ | $\mathbf{\approx 2,000}$ | $\mathbf{\approx 5,500}$ | $\mathbf{\approx 40,000}$ | $\mathbf{\approx 36,000}$ | $\mathbf{\approx 5,500}$ |
> >
> > In summary, HyperDQN is better than NoisyNet on 12 out of 18 tasks with 20M frames.
> >
> > Hope these results could help address your concern in **Comment 4**.

---

### Official Review · Reviewer_GmJU · 2021-11-07

**Correctness:** 3
**Technical Novelty And Significance:** 2
**Empirical Novelty And Significance:** 2
**Recommendation:** 6
**Confidence:** 3

**Main Review:**

The paper begins with a strong motivation towards computationally implementing RLSVI in Deep RL using a base and meta models, and also points out reasons why this is non-trivial. Also, the main differences between BootDQN and HyperDQN are also discussed, which is important because BootDQN is a highly relevant prior work in this domain. Overall the paper is well-written and has strong motivations that are relevant to the RL community at large.

The following are some of the questions based on my understanding of the paper and would be great if the authors could clarify some of them:

1. The meta-model in HyperDQN is considered as a linear parameterized model and there is a theoretical justification for this. What would happen to the empirical/theoretical results if the meta-model is a non-linear function of the latent variable z? I am inclined to think that the representation power for the posterior distribution would be better and thus should produce improvements overall.

2. How are the sigma_w and sigma_p parameters chosen in the objective function? Have the authors considered a baseline where an agent uses an identical artificial noise term for modifying the Q_{target} (i.e., sigma_w z^{\top} \epsilon_i ) to the one used in HyperDQN?

3. The Atari performance curves summarized in Fig 3 is not sufficient to demonstrate the utility of the presented approach. This curve is computed over all the Atari games (some of them require no exploration and some require a complex strategy for exploration). It would be more appropriate to show how the agent performs on “hard-exploration” and “easy-exploration” games (this classification is presented in Bellemare et al. 2016) rather than present a summary across all games.

Also, why are the agents only trained for 20M frames? It is conventional in Atari to run the agents for 200M frames (Machado et al. 2018). This makes it easier to compare the performance of HyperDQN wrto the prior methods in Atari.

A minor comment: why do some curves in Fig 3 start at 0 and another starts at a value lower than 0?

Bellemare, M., Srinivasan, S., Ostrovski, G., Schaul, T., Saxton, D., and Munos, R. Unifying count-based exploration and intrinsic motivation. In NeurIPS, 2016.

Machado, M. C., Bellemare, M. G., Talvitie, E., Veness, J., Hausknecht, M., & Bowling, M. (2018). Revisiting the arcade learning environment: Evaluation protocols and open problems for general agents. Journal of Artificial Intelligence Research, 61, 523-562.

4. How many random seeds were used to produce the learning curves in all the experiments (Atari, Mario and DeepSea)? It would be useful to add this information in the main text. Also, are the error bars reported in each of the presented learning curves?


**Summary Of The Paper:**

The paper presents HyperDQN as a practical algorithm of Randomized least-square value iteration (RLSVI).  HyperDQN consists of two parametric models: first is a base model that is similar to that of a DQN agent and the second a meta model that parameterizes the last layer of the Q-network as a function of a latent variable. The purpose of this architecture is to generate posterior samples using the diverse Q-value functions represented through this architecture. The paper demonstrates this approach empirically across a variety of Atari games, SuperMarioBros and DeepSea.


**Summary Of The Review:**

Overall, I think the contribution would be important to the RL community. My main concerns with the paper is that the presented approach claims to be helpful for exploration but the empirical results are not demonstrating this: One specific way to demonstrate this would be to show and emphasize that HyperDQN is able to achieve better returns on Atari games that are classified to be hard-exploration games. This would be a more powerful argument for HyperDQN.

---

> ### Author Response · Authors · 2021-11-16
> **Response to Reviewer GmJU (Part 1/2)**
>
> Thanks for your insightful review.
>
> **Question 1**: What would happen to the empirical/theoretical results if the meta-model is a non-linear function of the latent variable z?
>
> **Answer 1**: We agree that the representation power of the posterior distribution could be enhanced if the meta-model is a non-linear function. However, the negative effect is that the training problem becomes hard. We provide the empirical investigation of this direction in Appendix D.5. From Figure 7 in Appendix D.5, we observe that a non-linear hypermodel has a similar performance with the linear hypermodel. In particular, **it does not bring significant gains**. We believe the underlying reason is that a non-linear hypermodel is harder to train.
>
> ---
>
> **Question 2(a)**: How are the sigma_w and sigma_p parameters chosen in the objective function?
>
> **Answer 2(a)**: First, values of these parameters are listed in Table 2 in Appendix D. You can also read values from the following table.  Second, as stated in Remark 6 in Appendix D.3, we choose these values based on the concern of the parameter initialization. Third, to further address your concern, we provide ablation studies of these parameters in Appendix D.5. Results in Figure 6 indicate that our method is not sensitive to the noise scale $\sigma_\omega$ but a large prior scale $\sigma_p$ leads to poor performance since the posterior update is slow compared with the strong prior term.
>
> |                 | Atari | SuperMarioBros | Deep Sea |
> | --------------- | ----- | -------------- | -------- |
> | $\sigma_\omega$ | 0.01  | 0.01           | 0.01     |
> | $\sigma_p$      | 0.1   | 0.1            | 1.0      |
>
>
>
> **Question 2(b)**: Have the authors considered a baseline where an agent uses an identical artificial noise term for modifying the Q_{target} (i.e., sigma_w z^{\top} \epsilon_i ) to the one used in HyperDQN?
>
> **Answer 2(b)**: We are sorry that we do not fully understand your question. In our formulation, there is a noise $\sigma_\omega z^{\top} \xi$ in Equation (4.2) for HyperDQN, where $\xi$ is associated with each sample $(s, a, r, s^\prime)$. Correct me if wrong: do you mean to use the identical $\xi$ for all $(s, a, r, s^\prime)$ tuples? We did not try since it does not match our theory. It would be interesting to explore in the future.
>
> ---
>
> **Question 3**: The performance on "hard exploration" and "easy exploration" tasks. Achieving better returns on hard-exploration games would be a more powerful argument for HyperDQN.
>
> **Answer 3**: Thanks for this suggestion. In the revision, we show that the relative improvements of HyperDQN on "hard-exploration" and "easy-exploration" problems in Figure 11 and Figure 12 in Appendix E. In particular, we observe that HyperDQN has clear improvements in “easy exploration” environments (e.g., Battle Zone, Jamesbond, and Pong) and “hard exploration” environments (e.g., Frostbite, Gravitar, and Zaxxon) compared with BootDQN and OB2I. Unfortunately, HyperDQN does not work on Montezuma’s Revenge. In fact, all almost randomized exploration methods (including BootDQN and NoisyNet) cannot perform well on this task. One reason is that the extremely sparse reward provides limited feedback for feature selection, which is crucial for randomized exploration methods as argued in the introduction.
>
>
> In addition to the Atari suite, we also want to mention that SuperMarioBros-1-3-v1 and SuperMarioBros-2-2-v1 are two hard exploration problems due to sparse reward and long horizon. HyperDQN has significant improvements on these two tasks. In conclusion, HyperDQN can work on hard exploration tasks.
>
>
> For completeness, we would like to further explain why the metric in Figure 3 (i.e., averaged results over 56 tasks) is good to measure exploration efficiency. First, in practice, **before solving a task, we do not know whether this task is a "hard exploration" or "easy exploration" problem**. As a result, we hope an intelligent agent can perform well on both types of problems. Second, we want to highlight that **even for the "easy exploration" tasks, we still need more efficient strategies than epsilon-greedy**. Note that we are not saying that the hard exploration problem is less important. Instead, we need to pay attention to the improvement in all tasks.

---

> ### Author Response · Authors · 2021-11-16
> **Response to Reviewer GmJU (Part 2/2)**
>
> **Question 4**: Why are the agents only trained for 20M frames?
>
> **Answer 4**: Thanks for your comments. The main reason is that the 200M frames training leads to an unacceptable cost for us (also for most RL researchers), while 20M frames training is sufficient to show the exploration efficiency. We elaborate on this claim as follows.
>
> First, 200M frames training suffers heavy financial burden and time cost. Specifically, running DQN for 200M frames on a single environment takes about **30 days**. Unfortunately, we have limited servers so that we cannot run all experiments with 200M frames in an acceptable time. Furthermore, the experiment fee for such one experiment is about 30 * 24 * 0.5 = \$360 as the unit cost (of the server and power) for one hour is \\$0.5. Hence, the cost for 56 environments is about **\\$20,160** for a single algorithm, which we cannot afford at the current stage.
>
> Second, we would like to point out that the 20M frames training is commonly used in (Lee et al., 2019; Rashid et al. 2020, Bai et al., 2021). The main reason is that training with 20M frames is **sufficient to obtain near-optimal policies** in Atari (e.g., Battle Zone and Pong) and SuperMarioBros (e.g., 1-1 and 1-2).   In this sense, based on the current experiments, we believe HyperDQN will still be a strong baseline method after 200M.
>
>
> ---
>
> **Question 5**: Why do some curves in Fig 3 start at 0 and another starts at a value lower than 0?
>
> **Answer 5**: In Fig 3, Double DQN starts at a value lower than 0. This is because of the implementation in DQN Zoo. In particular, the evaluation policy of DoubleDQN at the initial stage is not a random policy so that its performance does not match 0. In contrast, other baselines can match a random policy due to randomization.
>
> ---
>
> **Question 6**: How many random seeds were used to produce the learning curves in all the experiments? Also, are the error bars reported in each of the presented learning curves?
>
> **Answer 6**: We use 3 random seeds for Atari and SuperMarioBros (due to limited computation resources) and 5 random seeds for Deep Sea. We have added these details in the main text in the revision. We have shown error bars for almost all figures. In particular, we do not show the error bar in Figure 3 since the error bar over 56 environments is large, which makes curves unclear. Note that in (O’Donoghue et al., 2018; Taiga et al., 2020), the error bar is also not shown for this type of figure. Please refer to Figure 16 and Figure 17 for learning curves of each environment, in which we clearly show the error bars.
>
> ---
>
> Lee, Su Young, et al. "Sample-efficient deep reinforcement learning via episodic backward update." NeurIPS 2019.
>
> Bai, Chenjia, et al. "Principled exploration via optimistic bootstrapping and backward induction." ICML, 2021.
>
> Rashid, Tabish, et al. "Optimistic exploration even with a pessimistic initialisation." ICLR 2020.
>
> Taiga, Adrien Ali, et al. "On Bonus-Based Exploration Methods in the Arcade Learning Environment." ICLR,  2020.
>
> O’Donoghue, Brendan, et al. "The uncertainty bellman equation and exploration." ICML, 2018.

---

### Author Response · Authors · 2021-11-16
**General Response to Reviewers**

We appreciate valuable comments from all reviewers. According to the suggestions from reviewers, we have revised our paper a lot. The modification parts are remarked in red in the revision. To get a quick overview of the revised paper, we summarize the main content as follows. **New 1, 2, 3 ...** stand for the new contents added in the revision.

- Introduction: illustrate challenges from extending RLSVI under deep RL scenario: feature learning and computation burden.
- Methodology: extend the hypermodel from bandit to deep RL.
    - (**New 1**) We explain the difference with (Dwaracherla et al., 2020): how the direct extension fails in deep RL
    - a theoretical result to explain why the (linear) hypermodel is effective to approximate the posterior distribution.
    - an objective function to address issues of extending RLSVI.
- Experiment: numerical results to validate the proposed method.
    - (**New 2**) HyperDQN could perform well on both hard exploration tasks and easy exploration tasks on Atari and SuperMarioBros.
    - (**New 3**) Validate HyperDQN is competitive to another strong baseline NoisyNet.
    - Explain and validate why commitment is important for HyperDQN.
    - Visualize the multi-step uncertainty and deep exploration of HyerDQN to provide insights.
- Conclusion: it's possible to extend HyperDQN for other domains.
    - (**New 4**) The extension HyperActorCritic (HAC) outperforms SAC on the hard exploration task Cart Pole.
    - Epistemic uncertainty qualification for offline RL.
    - Leverage the informative prior to accelerate exploration.

**New 1**: We explain the architecture difference with the original one in (Dwaracherla et al., 2020). Specifically, we illustrate the reason why the direct extension of (Dwaracherla et al., 2020) fails under the deep RL case. This helps clarify our novelty and contribution compared with (Dwaracherla et al., 2020).



**New 2**: We visualize the relative improvements on both "hard exploration" and "easy exploration" problems to better understand the advances.



**New 3**: We involve the discussion and empirical comparison with another baseline NoisyNet.



**New 4**: We add the result of the extension to continuous control tasks.



Accordingly, the appendix is also revised to include the following new contents:

- In Appendix D.5, we add extensive ablation studies on the parameter choices.
- In Appendix E.1, we show the figures of the relative improvements on both "hard exploration" and "easy exploration" problems.
- In Appendix F.2, we illustrate the trainability issue of HyperDQN.
- In Appendix F.4, we include the detailed discussion and comparison with NoisyNet.
- In Appendix F.5, we introduce the extension to continuous control tasks.
- In Appendix F.6, we present preliminary results when an informative prior function is available.

---

Dwaracherla, Vikranth, et al. "Hypermodels for exploration." ICLR 2020.

---

> ### Author Response · Authors · 2021-11-23
> **We have added more experiments**
>
> Dear all reviewers,
>
> We have added additional 6 experiments on Atari (alien, amidar, and battle zone) and SuperMarioBros (mario-3-1, mario-3-2, and mario-3-3) in Appendix F.4. Accordingly, Figure 22 and Figure 23 are updated. To summarize, HyperDQN is better than NoisyNet on 12 out of 18 benchmark tasks.  (We hope this could address the concern from Reviewer TPoj.)
>
> Thanks!

---

> ### Author Response · Authors · 2021-11-29
> **Looking Forward to Reviewer's Reply**
>
> We appreciate valuable comments from all reviewers. We believe we have carefully answered questions the reviewers asked about. To further address concerns, we have added experiments to support our claims. Since the final stage discussion is due soon, we sincerely look forward to feedback and possibly unsettled concerns from all reviewers. Also, we are glad to write a follow-up response if applicable.
>
> Thanks for your time and consideration!

---

### Comment · Area_Chair_BH85 · 2021-11-29
**Any final thoughts?**

Dear reviewers,

This is a follow-up on the private messages I sent through OpenReview. If you haven't already done so, please read the authors' response, and engage in the final discussions with them. At the very least, acknowledge their responses, and indicate whether they adequately addressed your concerns or not.

Today (November 29th) is the end of the discussion period.

Thank you,
Area Chair

---

### Public Comment · ~Haque_Ishfaq1 · 2022-02-06
**Relevant related work**

Dear Authors,

We just came across this nice paper. You may find this paper from ICML 2021 relevant: https://arxiv.org/abs/2106.07841

---

> ### Public Comment · ~Ziniu_Li1 · 2022-02-08
> **Thanks for sharing**
>
> Dear Haque,
>
> Thanks for sharing this paper. We will discuss this paper in the final version.

---

### Decision · Program_Chairs · 2022-01-20

**Decision:**

Accept (Poster)

**Comment:**

This paper extends Randomized least-square value iteration (RLSVI), which is a method for exploration-exploitation tradeoff that is suitable for linear FA, to the deep RL setting. A key component is using Hypermodels of Dwaracherla et al. (ICLR, 2020) to generate the weights of last layer of the DNN. This generates a learnable randomness required in an RLSVI-like procedure.
The paper provides some theoretical results regarding Hypermodels, and provides extensive experiments to show that their method is a competitive one in solving exploration problems.

The majority of the reviewers are positive about this work. The concerns include the incremental nature of this work and the empirical results. In my opinion, the algorithmic contribution is reasonable, but somehow incremental. The theoretical results are minor, but acceptable. The empirical results are extensive, though they have some shortcomings.
I explain the algorithmic and empirical contributions below:


**Algorithmic Contribution:**
Similar formulation has been done by  Dwaracherla et al. But that work does not consider the RL setting, and instead focuses on the bandit setting. This work provides such an extension. A straightforward application of Hypermodels does not work for DRL, but some simple, yet crucial, tricks needs to be applied to make it work. The trick is to use Hypermodel to generate the weights of the last layer, instead of all layers. Although this is simple, the fact that it enables the method to work for DRL paper is significant.


**Empirical Results:**
The empirical results are quite extensive. There are two main issues with them though:

a) Many experiments on the Atari Suite are terminated after 20M samples. This is shorter than usual.
b) The experiments are only repeated for 3 runs (seeds) -- except one, which used 5 runs.

The authors' answer for (a) is that they have a limited compute budget, and running for 200M samples would cost them about $20K. Also they argue that 20M samples is enough to show the benefit in a better exploration method, as shown by some other papers.

After some inquiries, it seems that this $20K value has the correct order to run the experiments on a cloud (maybe within a factor of 2 or 3). Given this prohibitive cost, I am willing to accept that 20M samples might be sufficient for proving the main points of this paper.  I am not giving a large weight to this in my evaluation.

The main concern for me, however, is having only 3 independent runs of the algorithms, especially given that the issue under study is the efficiency of exploration-exploitation tradeoff, and that the proposed method has a lot of randomness built-in. This is the main weakness of the empirical results in my opinion, and not the 20M samples issue.

For instance, Figure 3 (Human-normalized score over 56 environments in Atari 2600 suite) does not have any confidence interval information. And Figure 4 has some shaded areas around the curves, but it is not clear whether it is standard deviation, standard error, or some other quantification of uncertainty.

Even though this is a borderline paper, I recommend acceptance of this work under the expectation that the authors should improve their empirical studies. In particular, I recommend much larger number of independent runs (seeds), maybe around 10 or so, with proper information about the uncertainty of the estimates. If running this for all games is prohibitive, showing the performance with more runs on a subset of the games is sufficient. Also I encourage the authors to consider NeurIPS 2021 paper "Deep Reinforcement Learning at the Edge of the Statistical Precipice".